# MMLU-Pro: A More Robust and Challenging Multi-Task Language Understanding Benchmark

[1]Yubo Wang*, [1]Xueguang Ma*, [1]Ge Zhang, [1]Yuansheng Ni, [1]Abhranil Chandra,
[1]Shiguang Guo, [1]Weiming Ren, [1]Aaran Arulraj, [1]Xuan He, [1]Ziyan Jiang, [1]Tianle Li,
[1]Max Ku, [2]Kai Wang, [1]Alex Zhuang, [1]Rongqi Fan, [3]Xiang Yue, [1]Wenhu Chen*

[1]University of Waterloo, [2]University of Toronto, [3]Carnegie Mellon University

## Abstract

In the age of large-scale language models, benchmarks like the Massive Multitask Language Understanding (MMLU) have been pivotal in pushing the boundaries of what AI can achieve in language comprehension and reasoning across diverse domains. However, as models continue to improve, their performance on these benchmarks has begun to plateau, making it increasingly difficult to discern differences in model capabilities. This paper introduces MMLU-Pro, an enhanced dataset designed to extend the mostly knowledge-driven MMLU benchmark by integrating more challenging, reasoning-focused questions and expanding the choice set from four to ten options. Additionally, MMLU-Pro eliminates the trivial and noisy questions in MMLU. Our experimental results show that MMLU-Pro not only raises the challenge, causing a significant drop in accuracy by 16% to 33% compared to MMLU but also demonstrates greater stability under varying prompts. With 24 different prompt styles tested, the sensitivity of model scores to prompt variations decreased from 4-5% in MMLU to just 2% in MMLU-Pro. Additionally, we found that models utilizing Chain of Thought (CoT) reasoning achieved better performance on MMLU-Pro compared to direct answering, which is in stark contrast to the findings on the original MMLU, indicating that MMLU-Pro includes more complex reasoning questions. Our assessments confirm that MMLU-Pro is a more discriminative benchmark to better track progress in the field.

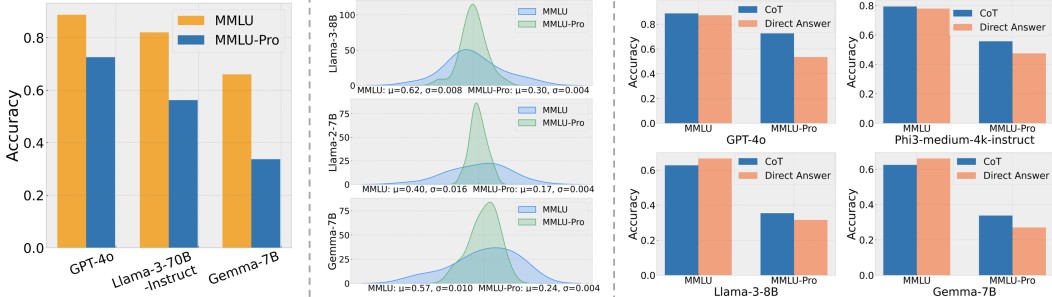

Figure 1: Comparing between MMLU and MMLU-Pro: (Left) Performance gap; (Center) Accuracy distributions affected by 24 prompts, with taller and thinner profiles indicating more stability and shorter and wider profiles indicating greater fluctuations; (Right) Performance using CoT vs. Direct.

*Core Contributors. ✉: {y726wang, x93ma, wenhuchen}@uwaterloo.ca

38th Conference on Neural Information Processing Systems (NeurIPS 2024) Track on Datasets and Benchmarks.

# 1 Introduction

In recent years, advancements in large language models (LLMs) have significantly transformed the field of natural language processing (NLP). These models, including state-of-the-art examples like GPT-4, Gemini, and Claude [2, 30, 6], are pushing the envelope both in general applicability across various tasks and specialized performance in specific areas. A key objective in this ongoing development is achieving expert-level intelligence, characterized by performance that meets or surpasses the top 10% of skilled adults in a diverse range of tasks [26].

To effectively track the progress towards the goal of this expert-level intelligence, it is essential to evaluate these models on a broad range of tasks. There are multiple popular benchmarks used to measure such general intelligence. For example, AGIEval [48] focuses on general-exam questions from SAT, Gaokao, GRE, etc. ARC [12] focuses on science-based questions. BBH [33] focuses on solving hard synthetic tasks. MMLU [18] includes a broad range of exam questions from 57 subjects across STEM, the humanities, the social sciences, etc. Among these benchmarks, MMLU emerged as the de facto standard for evaluating LLMs due to its broad coverage and high quality. However, the rapid progress of current LLMs has quickly led to performance saturation on MMLU. Since GPT-4 achieved 86.4% in March 2023, there has not been any significant progress on the benchmark. Most recent frontier models like GPT-4-Turbo, Gemini-1.5-Pro, Claude, and LLaMA-3-400B (all published in early-mid 2024) all settle at an accuracy between 86% - 87%. The recent published GPT-4o has achieved remarkable performance boost (10+% improvement) on MATH [19], Chatbot Arena [11]. However, it only obtains 1% improvement on MMLU to obtain 87.4%. This leads us to re-examine the effectiveness of MMLU in measuring future (stronger) LLMs. Besides the saturation issue, the performance on MMLU is also known to be highly sensitive to the prompt and scoring function [47, 3], which causes significant order changes in the leaderboard. Here, we conjecture that these issues are due to the following causes:

1. The questions in MMLU only have three distractor options. LLMs could potentially exploit shortcuts to derive the answer without truly understanding the rationale. This could lead to an overestimate of LLMs' true performance, also leading to a degree of instability.

2. The questions in MMLU are mostly knowledge-driven without requiring too much reasoning, especially in the STEM subjects, which reduces its difficulty. In fact, most models achieve better performance with 'direct' answer prediction without chain-of-thought [41].

3. There is a portion of questions that are either unanswerable or mistakenly annotated. This dataset noise leads to a lower ceiling, which the frontier models hit.

These issues have highlighted the need for more challenging, discriminative, and reliable datasets to track the progress of LLMs. In this paper, we introduce MMLU-Pro: a comprehensive benchmark designed for proficient-level multi-discipline language understanding and reasoning. MMLU-Pro spans 14 diverse domains including mathematics, physics, chemistry, law, engineering, psychology, and health, encompassing over 12,000 questions and thus meeting the breadth requirement. MMLU-Pro is distinctive from MMLU in the following aspects:

1. MMLU-Pro has ten options, which contain 3x more distractors than MMLU. By increasing the distractor numbers, we significantly reduce the probability of correct guess by chance to boost the benchmark's difficulty and robustness.

2. MMLU-Pro increases the portion of challenging college-level exam problems. These questions require LLM to perform deliberate reasoning in different domains to derive the final answer.

3. We integrate two rounds of expert reviews to reduce the noise of the dataset. The first round is based on expert verification. In the second round, we utilize the SoTA LLMs to identify potential errors and employ annotators to perform more targeted verification.

We evaluated more than 50 LLMs including open-source and closed-source models, such as GPT-4o [17], Claude-3-Opus [6], and Gemini [30], LLaMA-3 [35], Phi-3 [1] on MMLU-Pro. Our key findings are summarized as follows:

1. MMLU-Pro presents significant challenges; notably, the leading model, GPT-4o, only achieves an accuracy of 72.6% and GPT-4-Turbo reaches 63.7%, indicating substantial room for improvement.

2. MMLU-Pro is more discriminative than MMLU in distinguishing the nuances between models. For example, the gap between GPT-4o and GPT-4-Turbo is 1% on MMLU, while it becomes 9% on MMLU-Pro. This discriminative nature makes MMLU-Pro a more suitable benchmark.

3. Advanced open-source models like Llama-3-70B-Instruct [24] and DeepSeek-V2-Chat [15], while not yet performing at the level of leading closed-source models such as GPT-4o and Claude-3-Opus, have shown performance that is close to Claude-3-Sonnet.

4. MMLU-Pro necessitates chain-of-thought (CoT) [41] to achieve promising results. For instance, CoT can boost the performance of GPT-4o by 19%. In contrast, CoT will actually hurt the performance of models on MMLU. This reflects the necessity to perform deliberate reasoning on MMLU-Pro, which is not required in the knowledge-driven MMLU questions.

5. Our error analysis on 120 erroneous cases of GPT-4o, the current top-performing model, reveals that 39% of errors are due to flaws in the reasoning process, 35% stem from a lack of specific domain expertise, and another 12% from computational errors. These results highlight the MMLU-Pro benchmark's difficulties and indicate areas needing further research and model enhancement.

## 2 Related Work

### 2.1 Large Language Models

Recent advancements in LLMs have significantly propelled the field of natural language processing. GPT-3 [7] demonstrated robust few-shot prediction capabilities, interpreting tasks and examples from natural language inputs. Subsequent models like InstructGPT [28], which employ human-feedback reinforcement learning, have achieved strong user instruction-following capability. More recent models including GPT-4o, GPT-4, Claude-3, Gemini, and Llama-3, have shown notable improvements in complex reasoning across various domains. To rigorously assess and push the capabilities of these LLMs, we introduce MMLU-Pro, a new benchmark designed to test the upper limits of reasoning and knowledge in advanced language models.

### 2.2 LLMs Evaluation Benchmarks

In recent years, the development of various benchmarks has significantly enhanced the evaluation of Large Language Models (LLMs). For instance, GLUE [38] and its successor SuperGLUE [37], have played a pivotal role in advancing language understanding tasks, setting the stage for more specialized evaluations. Other recent benchmarks, including MMLU [18], HELM [22], BigBench [32], HellaSwag [45], and the AI2 Reasoning Challenge (ARC) [12], have broadened the scope by assessing capabilities across language generation, knowledge understanding, and complex reasoning [9].

To facilitate performance comparison across diverse LLMs, several leaderboards have been established, such as the OpenLLM Leaderboard [27] and OpenCompass [14]. However, as LLM capabilities have rapidly advanced, leaderboard scores have become increasingly concentrated at the top, with models like GPT-4 achieving near-perfect scores on multiple benchmarks. This trend highlights the urgent need for more challenging benchmarks to fully test the limits of LLM capabilities.

Recent studies have revealed that the performance of Large Language Models (LLMs) on current benchmarks is not robust to minor perturbations [25, 31]. Specifically, slight variations in the style or phrasing of prompts can lead to significant shifts in model scores. Beyond the inherent non-robustness of the models themselves, the typical four-option format of multiple-choice questions (MCQs) also contributes to this instability in model scoring. This format may not sufficiently challenge the models or differentiate between closely performing systems, leading to potential overestimations of their capabilities. Our new benchmark, MMLU-Pro, aims to address these issues by introducing more complex questions and increasing the number of answer choices from four to ten, thus enhancing performance differentiation and reducing the likelihood of correct guesses by chance.

## 3 The MMLU-Pro Benchmark

### 3.1 Overview

Our dataset comprises 14 discipline subsets, totaling 12,032 questions, with their distribution and origins detailed in Figure 3. It integrates questions from several sources: (1) Original MMLU

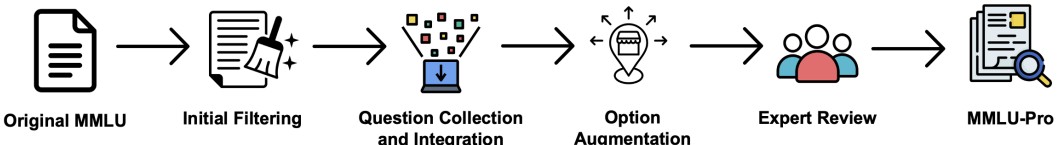

Figure 2: MMLU-Pro Dataset Construction Pipeline

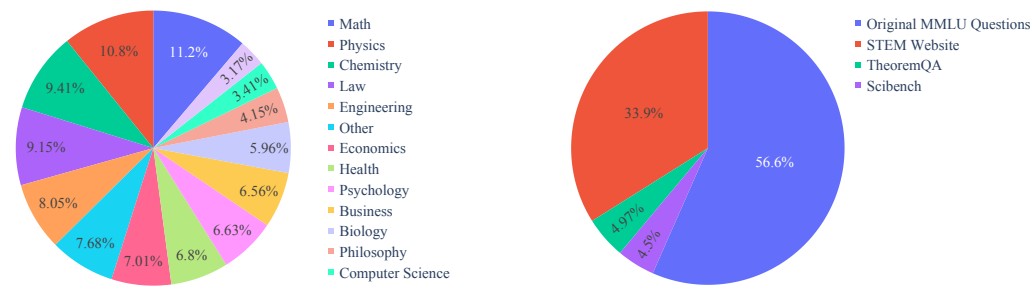

(a) Distribution of Disciplines in MMLU-Pro      (b) Data Source Distribution in MMLU-Pro

Figure 3: Distribution of disciplines and question sources in the MMLU-Pro dataset

Questions, which form the core of our dataset and include questions adapted from the original MMLU dataset with trivial and erroneous questions removed; (2) the STEM Website, providing selected high-quality STEM problems from online platforms; (3) TheoremQA, featuring high-quality, human-annotated questions that necessitate the application of theorems for resolution; and (4) SciBench, which includes advanced science questions derived from college exams, ensuring the inclusion of curriculum-aligned questions. For more details about detailed data statistics, prompt instructions, and incorrect answer cases we eliminated, please refer to Appendix A.1.

## 3.2 Dataset Construction Pipeline

**Initial Filtering** As shown in Figure 2, our dataset construction begins with a comprehensive review of the original MMLU dataset, which we streamline by merging the previous 57 subject categories into 14 broader categories. This restructuring aims to better focus the evaluation on key areas of knowledge and reduce redundancy. We also aim to eliminate overly simple questions that fail to challenge the models effectively. We employ eight models: Llama-2-7B, Llama-2-7B-Chat, Llama-2-13B, Llama-2-13B-Chat [35], Mistral-7B [20], Gemma-7B [34], Yi-6B, and Yi-6B-Chat [43]. Each model is evaluated on the MMLU, and questions answered correctly by more than four models are considered as "too easy" and subsequently excluded from consideration. Using this criterion, a total of 5,886 questions are filtered out across the various subjects.

**Question Collection and Integration** We then expand our dataset by incorporating questions from the STEM Website [2], TheoremQA [10], and SciBench [40]. It is important to note that the questions from the STEM Website are in the form of problem statements with solutions, while those from TheoremQA are in the format of questions accompanied by brief answers. To adapt these for our dataset, we utilize GPT-4-Turbo[3] to extract short answers from the solutions, serving as the correct answer options. We also generate three additional distractors for each question. We then compare the solutions with the extracted answers manually, to remove questions where the extracted answers are incomplete or incorrect. This step is essential for aligning the STEM Website and TheoremQA questions with those from other sources, preparing them for future option augmentation.

**Option Augmentation** We enhance the question options from four to ten using GPT-4-Turbo, introducing six additional choices. These are not merely quantitative additions; rather, they serve as plausible distractors that necessitate discerning reasoning for correct selection. This approach

---

[2]`https://stemez.com/subjects`

[3]In this paper, GPT-4-Turbo refers to GPT-4-turbo-2024-04-09. For more details, see `https://platform.openai.com/docs/models/gpt-4-turbo-and-gpt-4`.

Table 1: Distribution of Issues Identified during the Expert Review Process

|                       | MMLU | TheoremQA | SciBench | STEM Website |
|-----------------------|------|-----------|----------|--------------|
| Incorrect Answer      | 350  | 0         | 11       | 483          |
| False Negative Options| 1953 | 5         | 15       | 293          |
| Bad Question          | 385  | 1         | 15       | 862          |

significantly lowers the likelihood of correctly guessing an answer, thereby increasing both the difficulty and the robustness of the benchmark. In experiments, we found that GPT-4-Turbo does not gain additional advantage from such an augmentation procedure.

**Expert Review**    The expert review process for our dataset construction comprises two main phases to ensure its quality and reliability. **Phase 1: Verification of Correctness and Appropriateness** involves experts verifying the accuracy of each answer, removing questions unsuitable for a multiple-choice format, and discarding questions that lack necessary information or require non-textual elements like images or tables. **Phase 2: Ensuring Distractor Validity** involves the Gemini-1.5-Pro model re-evaluating all answer options to identify potential false negatives. In this context, a 'false negative' refers to a correct option initially misclassified as incorrect. Subsequently, human experts rigorously review these identified options to ensure that actual distractors are indeed incorrect and distinctly different from the correct answer. In Table 1, we present a distribution of issues identified during the expert review process. For better illustration, we categorize them into three types:

- **Incorrect Answers:** refer to instances where the provided answer is incorrect. There are two main sources of these errors: the pre-existing errors within the original MMLU dataset and errors on the STEM Website arising from flawed or incomplete answer extraction.
- **False Negative Options:** primarily arise from distractors generated in two key stages: converting single answers into four options from sources like the STEM Website and TheoremQA, and further expanding these four options to ten in the option augmentation phase. Human experts will remove each False Negative Option, keeping the correct answer and suitable distractors. In our dataset, 83% have ten options, 17% have fewer, and the average options count per question is 9.47.
- **Bad Questions:** include several problematic aspects: (1) Questions that require non-text information such as images or tables. (2) Questions that lack sufficient textual information to derive a conclusive answer. (3) Questions that are unsuitable for a multiple-choice format, such as proof problems, true or false questions, and open-ended questions.

# 4   Experimental Setup

**Few-Shot Chain-of-Thought Prompting**

We utilize a 5-shot Chain-of-Thought (CoT) [41] approach to measure model performance on challenging tasks presented by MMLU-Pro. This CoT reasoning, adapted from Chain-of-Thought Hub [16], incorporates essential reasoning steps from the original MMLU dataset. Our approach introduces two enhancements: firstly, extending the original options available from the Chain-of-Thought Hub, and secondly, selecting five representative demonstration examples for each discipline. Unlike traditional performance measures such as Perplexity (PPL) which primarily focus on linguistic probabilities, our method emphasizes reasoning capabilities, crucial for handling the complexities of MMLU-Pro. A comprehensive comparison of performances using direct answering and CoT methods will be presented in Section 6.2, demonstrating the effectiveness of the CoT approach.

**Answer Extraction**

To extract answers from the model-generated reasoning content in response to the MMLU-Pro dataset, we initially use the regular expression `answer is \(?\([A-J]\)?\)` to match the format specified in the prompt instructions and few-shot examples. If this regex fails to retrieve a valid response, possibly due to formatting deviations by the model, we employ a secondary regex `\.*\[aA\]nswer:\s*\([A-J]\)` for a second attempt to extract the answer. If both of them fail to retrieve a valid response, a fallback mechanism is implemented where a random option from the answer choices is selected. This ensures consistent answer provision across all evaluations.

Table 2: Models Performance on MMLU-Pro, CoT. Values are accuracies in percentages. (All the models use 5 shots except Gemini-1.5-pro and Gemini-1.5-Flash, which use 0 shots.)

| Models | Overall | Math | Physics | Engineering | History | Law | Psychology |
|---|---|---|---|---|---|---|---|
| Closed-source Models | | | | | | | |
| GPT-4o [17] | 72.6 | 76.1 | 74.7 | 55.0 | 70.1 | 51.0 | 79.2 |
| Gemini-1.5-Pro [30] | 69.0 | 72.8 | 70.4 | 48.7 | 65.6 | 50.8 | 77.2 |
| Claude-3-Opus [13] | 68.5 | 69.6 | 69.7 | 48.4 | 61.4 | 53.5 | 76.3 |
| GPT-4-Turbo [2] | 63.7 | 62.8 | 61.0 | 35.9 | 67.7 | 51.2 | 78.3 |
| Gemini-1.5-Flash [30] | 59.1 | 59.6 | 61.2 | 44.2 | 53.8 | 37.3 | 70.1 |
| Yi-large [23] | 58.1 | 64.8 | 57.0 | 45.4 | 49.6 | 36.2 | 50.6 |
| Claude-3-Sonnet [13] | 56.8 | 49.0 | 53.1 | 40.5 | 57.2 | 42.7 | 72.2 |
| Open-source Models | | | | | | | |
| Llama-3-70B-Instruct [24] | 56.2 | 54.0 | 49.6 | 43.6 | 56.9 | 39.9 | 70.2 |
| Phi-3-medium-4k-instruct [1] | 55.7 | 52.2 | 49.4 | 37.9 | 57.2 | 38.3 | 73.4 |
| DeepSeek-V2-Chat[15] | 54.8 | 53.7 | 54.0 | 31.9 | 45.3 | 40.6 | 66.2 |
| Llama-3-70B [24] | 52.8 | 49.7 | 49.8 | 35.0 | 57.7 | 35.0 | 71.4 |
| Qwen1.5-72B-Chat [5] | 52.6 | 52.3 | 44.2 | 36.6 | 55.9 | 38.5 | 67.7 |
| Yi-1.5-34B-Chat [43] | 52.3 | 56.2 | 49.4 | 34.4 | 52.8 | 34.8 | 64.3 |
| MAmmoTH2-8x7B-Plus[44] | 50.4 | 50.3 | 45.7 | 34.0 | 50.9 | 35.5 | 63.8 |
| Qwen1.5-110B [5] | 49.9 | 50.4 | 41.4 | 35.3 | 54.1 | 35.1 | 66.3 |
| Phi-3-mini-4k-instruct [1] | 45.7 | 41.8 | 41.0 | 28.7 | 41.5 | 28.5 | 65.2 |
| Mixtral-8x7B-Instruct-v0.1 [21] | 43.3 | 36.3 | 39.9 | 29.2 | 44.6 | 32.1 | 63.4 |
| Yi-34B [43] | 43.0 | 31.8 | 35.0 | 32.6 | 52.0 | 32.7 | 62.5 |
| Mixtral-8x7B-v0.1 [21] | 41.0 | 34.1 | 37.2 | 27.9 | 47.5 | 27.1 | 61.0 |
| Llama-3-8B-Instruct [24] | 41.0 | 36.1 | 34.4 | 31.3 | 42.3 | 26.5 | 59.4 |
| Staring-7B [49] | 37.9 | 34.9 | 38.5 | 27.0 | 43.6 | 24.7 | 32.5 |
| c4ai-command-r-v01 [8] | 37.9 | 26.3 | 28.3 | 24.8 | 47.5 | 34.0 | 58.5 |
| Llama-2-70B [35] | 37.5 | 26.8 | 28.2 | 23.5 | 45.9 | 28.6 | 59.0 |
| OpenChat-3.5-8B [39] | 37.2 | 36.2 | 30.5 | 26.9 | 39.9 | 24.6 | 54.5 |
| InternMath-20B-Plus [42] | 37.1 | 56.1 | 24.0 | 30.4 | 20.5 | 15.2 | 42.3 |
| Llama3-Smaug-8B [29] | 36.9 | 33.2 | 37.3 | 19.8 | 42.0 | 26.5 | 28.6 |
| Llama-3-8B [24] | 35.4 | 30.4 | 31.4 | 25.5 | 36.2 | 19.6 | 53.3 |
| Gemma-7B [34] | 33.7 | 25.1 | 27.6 | 22.7 | 36.8 | 21.7 | 51.8 |
| InternMath-7B-Plus [42] | 33.5 | 48.3 | 22.8 | 28.2 | 19.2 | 14.4 | 38.3 |
| Zephyr-7B-Beta [36] | 33.0 | 23.6 | 35.7 | 23.9 | 32.0 | 22.0 | 28.2 |
| Mistral-7B-v0.1 [20] | 30.9 | 23.5 | 24.8 | 22.4 | 32.6 | 20.7 | 48.9 |
| Neo-7B-Instruct [46] | 28.7 | 35.5 | 23.7 | 19.1 | 28.2 | 18.0 | 36.2 |
| Llemma-7B [4] | 23.5 | 21.6 | 25.7 | 23.8 | 15.2 | 14.8 | 29.6 |
| Gemma-2B [34] | 15.9 | 16.3 | 15.6 | 12.7 | 15.4 | 12.3 | 16.1 |

# 5 Results and Analysis

Table 2 showcases the performance of frontier models on the MMLU-Pro benchmark. Due to space constraints, we selected a subset of models and representative domains, including three reasoning-focused subjects (Mathematics, Physics, Engineering) and three knowledge-heavy subjects (History, Law, Psychology). Full results are available on our leaderboard [4].

## 5.1 Overall Performance

GPT-4o [17] emerges as the strongest model with an overall accuracy of 72.6%, showing superior performance across all subjects. Additionally, Phi-3-medium-4k-instruct (14B parameters) and Phi-3-mini-4k-instruct (3.8B parameters) perform exceptionally well, possibly due to their pre-training on high-quality educational data and code.

Additionally, Results from Table 2 indicate that top-tier closed-source models outperform the open-source ones. Among the leading open-source models, Llama-3-70B-Instruct performs the best, achieving an accuracy of 56.2%, close to that of Yi-Large and Claude-3-Sonnet. However, it still significantly lags behind GPT-4o and Claude-3-Opus in all subjects.

---

[4]Please visit our leaderboard at `https://huggingface.co/spaces/TIGER-Lab/MMLU-Pro`.

## 5.2 Subject-Specific Insights

**Math and Physics:** In computation and reasoning-intensive subjects like Math and Physics, we observe significant performance disparities among models. The gap stretches from over 70% accuracy for GPT-4o to just over 20% for Mistral-7B-v0.1, illustrating a wide range in capabilities and underscoring the value of our benchmark in distinguishing these differences.

**History and Psychology:** In knowledge-intensive subjects such as History and Psychology, models generally show a higher performance floor compared to reasoning-intensive disciplines. Interestingly, the DeepSeek-V2-Chat model underperforms relative to its peers in these subjects, indicating its comparatively stronger reasoning abilities over its knowledge retrieval capabilities.

**Engineering and Law:** Among the 14 subjects evaluated, Engineering and Law consistently scored lower. Upon reviewing model outputs, we found that the lower scores in Engineering are largely due to the addition of new questions sourced from the STEM Website, which require complex formula derivations and multi-step calculations. This aspect leaves substantial room for improvement in future, more advanced models. Law scores suffer as questions become more intricate and detailed with additional options, necessitating a deeper understanding of legal reasoning.

## 5.3 Error Analysis

In this section, we explore an error analysis of GPT-4o, currently the best-performing model on the MMLU-Pro benchmark, to examine its performance strengths and weaknesses. This examination not only highlights areas where the model falls short but also provides insights that could inform future improvements in both its architecture and training processes. We conducted a detailed review of 120 randomly selected erroneous predictions made by GPT-4o. These errors were analyzed by expert annotators who determined the underlying causes of each misprediction using their expert judgment. Specific cases and further detailed discussions are provided in Appendix A.6.

**Reasoning Errors (39%)** The model frequently encounters difficulties with logical reasoning, even when it recalls the correct information and knowledge. These issues often arise from logical inconsistencies in its responses, likely due to its dependence on recognizing patterns in training data rather than engaging in a true understanding of the problem.

**Lack of Specific Knowledge (35%)** A fundamental root cause of domain-specific errors in the GPT-4o model is the lack of specialized knowledge. Errors such as incorrect financial calculations and misapplications of optical principles highlight this issue.

**Calculation Errors (12%)** We distinguish calculation errors from reasoning errors to aid model developers, as many AI systems can effectively utilize calculators or Python for complex, multi-step calculations. In our review of error cases, it is common to find instances where the model has the correct formula but makes errors in computing values.

**Other Errors** The remaining errors include No Selection Made (5%), Question Understanding Errors (4%), Generation Issues (2%), Annotation Errors (2%), and Answer Extraction Errors (1%). These errors are attributed to various factors, such as limitations in final response selection, complex text interpretation challenges, limitations in response generation, inaccuracies in data annotation, and issues in extracting precise answers from model outputs.

# 6 Comparison with MMLU

In this section, we will compare the MMLU and MMLU-Pro benchmarks from three perspectives: difficulty level, reasoning strength, and robustness degree.

## 6.1 Difficulty Level

In Figure 4, we present scores of different models on both MMLU and MMLU-Pro benchmarks. It is evident that as language model capabilities enhance, the scores on MMLU are not only increasing but also clustering closely together, making it difficult to distinguish between models. For instance,

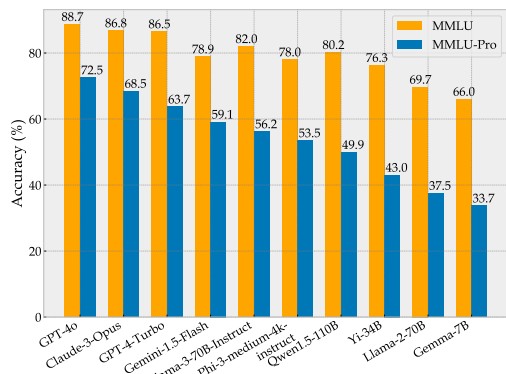

Figure 4: Performance Comparison: MMLU vs. MMLU-Pro

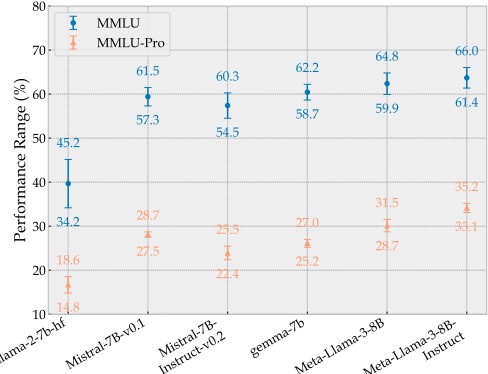

Figure 5: Performance Variability under Different Prompts on MMLU and MMLU-Pro

Table 3: Accuracy Differences of Models Between CoT and Direct Answering (%)

| Model Name | MMLU | | | MMLU-Pro | | |
|---|---|---|---|---|---|---|
| | CoT | Direct Answer | CoT - DA | CoT | Direct Answer | CoT - DA |
| GPT-4o | 88.7 | 87.2 | **1.5** | 72.6 | 53.5 | **19.1** |
| GPT-4-Turbo | 86.5 | 86.7 | **-0.2** | 63.7 | 48.4 | **15.3** |
| Phi3-medium-4k-instruct | 79.4 | 78.0 | **1.4** | 55.7 | 47.5 | **8.2** |
| Llama-3-8B | 62.7 | 66.6 | **-3.9** | 35.4 | 31.5 | **3.9** |
| Gemma-7B | 62.4 | 66.0 | **-3.6** | 33.7 | 27.0 | **6.7** |

models like Gemini-1.5-Flash, Llama-3-70B-Instruct, Phi-3-medium-4k-instruct, and Qwen1.5-110B all score between 78% and 82%, a narrow 4% range that encompasses four models, challenging the differentiation of their performance. MMLU-Pro expands this range to approximately 10%. Similarly, the score difference between models like GPT-4o, Claude-3-Opus, and GPT-4-Turbo has widened from about 2% on MMLU to around 9% on MMLU-Pro. Additionally, the increased difficulty in MMLU-Pro ensures ample room for future model improvement. Currently, the best-performing model, GPT-4o, scores 72.6% on MMLU-Pro, leaving a substantial margin of 27.4% for potential improvement, whereas MMLU offers only about 11.3% space for further enhancement.

## 6.2 Reasoning Level

According to Table 3, we can observe differences in performance between the Chain of Thought (CoT) method and Direct Answering (DA) across various models on MMLU and MMLU-Pro. The comparison shows that the CoT method generally results in more significant performance improvements on MMLU-Pro compared to MMLU.

Specifically, GPT-4o improves by 1.5% using the Chain of Thought (CoT) method compared to direct answering on MMLU, while on MMLU-Pro, its improvement reaches 19.1%. Similarly, GPT-4-Turbo shows a 15.3% increase in performance using CoT over direct answering on MMLU-Pro, although its performance slightly decreases by 0.2% on MMLU. Other models such as Phi3-medium-4k-instruct, Llama-3-8B, and Gemma-7B also display similar trends, exhibiting greater performance improvements using CoT on MMLU-Pro compared to direct answering. These findings indicate that the MMLU-Pro benchmark is specifically designed to assess deeper and more complex reasoning skills, as evidenced by the enhanced performance of models using chain-of-thought (CoT), highlighting its focus on professional-level problem-solving.

## 6.3 Robustness Degree

It is widely recognized that even minor variations in prompts can significantly impact model outputs, leading to substantial fluctuations when evaluating models. This poses challenges for accurately ranking models and maintaining consistent leaderboards [3]. This sensitivity is generally attributed to

models' lack of robustness [47], a characteristic tied to the underlying principles of language models that fall outside the scope of this study. However, a high-quality benchmark should aim to minimize the impact of prompt variability on scores, ensuring more consistent and reliable evaluations.

To assess this, we evaluated models using 24 different but reasonable prompts. Figure 5 showcases the score range for different models under varying prompts. On the MMLU benchmark, the influence of these prompts generally ranges between 4-5%, with peaks up to 10.98%. In contrast, on the MMLU-Pro benchmark, the impact of prompt changes is generally around 2%, with a maximum of 3.74%. This reduced variability highlights an improvement in consistency and reliability over the original MMLU benchmark, ensuring more reliable assessments of language models' capabilities.

## 7 Limitations

The MMLU-Pro dataset, while enhancing the complexity of MMLU by incorporating more challenging, reasoning-focused questions, remains constrained by the limitations of the multiple-choice format. This format may not capture the depth of comprehension and creative response generation as effectively as open-ended answers, which better reflect real-world scenarios. Additionally, MMLU-Pro exclusively focuses on language models and does not include assessments for multi-modal models, limiting its applicability in scenarios requiring synthesis of visual, auditory, and textual data.

## 8 Conclusion

In this paper, we introduce MMLU-Pro, a more challenging benchmark designed to elevate the assessment of multi-task language understanding capabilities in language models. By incorporating more complex, reasoning-intensive tasks, MMLU-Pro addresses the performance saturation observed in previous benchmarks, effectively differentiating models' capabilities. Our evaluations show that even leading models like GPT-4o encounter significant challenges, indicating a successful increase in difficulty and an improvement in the benchmark's ability to test deeper cognitive processes. MMLU-Pro also enhances its robustness by reducing dependency on prompt styles, making it a valuable tool for advancing our understanding of AI language capabilities. As AI technology evolves, we hope MMLU-Pro plays a crucial role in pushing the boundaries of what language models can achieve.

## Acknowledgments and Disclosure of Funding

We would like to thank Reddit user Dorrin Verrakai, who provided invaluable feedback for this work. We also express our gratitude to Ankesh Anand from Google DeepMind and Ning Shang from Microsoft for their insightful comments and suggestions. Additionally, we appreciate the contributions of all open-source language model providers, whose efforts have significantly propelled the advancement of research in this field.

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

# A  Appendix

## A.1  Dataset Construction Details

**Initial Filtering Details**    Table 4 provides a comprehensive overview of the initial data filtering performed on the Massive Multitask Language Understanding (MMLU) dataset across various academic disciplines. It details the original number of items in each discipline, the number filtered out due to specific criteria, the percentage of items filtered, and the remaining number of items after filtering. Among the disciplines, Business, History, Other, and Psychology exhibit the highest filtering percentages, with more than 50% of the original questions being filtered out. This indicates that the questions in these disciplines are relatively simple and that many current language models have already mastered the relevant knowledge.

Table 4: Summary of Initial Filtering on MMLU Across Various Disciplines

| Discipline | Original Number | Filtered Out | Percentage Filtered | Remaining |
|---|---|---|---|---|
| Biology | 454 | 225 | 49.56% | 229 |
| Business | 437 | 263 | 60.18% | 174 |
| Chemistry | 303 | 81 | 26.73% | 222 |
| Computer Science | 412 | 127 | 30.83% | 285 |
| Economics | 742 | 275 | 37.06% | 467 |
| Engineering | 145 | 55 | 37.93% | 90 |
| Health | 1562 | 681 | 43.60% | 881 |
| History | 930 | 526 | 56.56% | 404 |
| Law | 1763 | 623 | 35.34% | 1140 |
| Math | 1063 | 172 | 16.18% | 891 |
| Other | 2340 | 1301 | 55.60% | 1039 |
| Philosophy | 2012 | 764 | 37.97% | 1248 |
| Physics | 629 | 169 | 26.87% | 460 |
| Psychology | 1145 | 624 | 54.50% | 521 |
| Total | 13937 | 5886 | 42.23% | 8051 |

**Distribution of MMLU-Pro Question Origin Details**    Table 5 illustrates the varied sources of questions across different disciplines, highlighting the dependency on specific databases. Disciplines like Law, Other, Health, Philosophy, and History rely exclusively on questions from MMLU. Engineering, on the other hand, predominantly uses questions from STEM websites, accounting for 93.08% of its total. Similarly, Business, Chemistry, and Biology show a significant reliance on STEM website sources. Additionally, Math and Physics display a more diversified sourcing pattern, relatively evenly drawing questions from three or four sources.

**Details of LLMs Utilization in Dataset Construction**    In Table 6, we showcase the prompts used with Large Language Models (LLMs) in the dataset construction process. These prompts include using GPT-4-Turbo to convert problems from STEM Website and TheoremQA into multiple-choice questions (MCQs), expanding four-option MCQs to ten-option MCQs with GPT-4-Turbo, and employing Gemini-1.5-Pro to recall False Negative Options.

Table 5: Distribution of Question Origin Across Different Disciplines. (The percentages in parentheses represent the proportion of each source relative to the total number of questions in that discipline.)

| Disciplines | Total Number | MMLU | STEM Website | TheoremQA | Scibench |
|---|---|---|---|---|---|
| Math | 1351 | 846 (62.66%) | 0 | 344 (25.46%) | 161 (11.92%) |
| Physics | 1299 | 411 (31.64%) | 617 (47.50%) | 104 (8.01%) | 167 (12.86%) |
| Chemistry | 1132 | 178 (15.72%) | 741 (65.46%) | 0 | 213 (18.82%) |
| Law | 1101 | 1101 (100%) | 0 | 0 | 0 |
| Engineering | 969 | 67 (6.92%) | 902 (93.08%) | 0 | 0 |
| Other | 924 | 924 (100%) | 0 | 0 | 0 |
| Economics | 844 | 444 (52.61%) | 400 (47.39%) | 0 | 0 |
| Health | 818 | 818 (100%) | 0 | 0 | 0 |
| Psychology | 798 | 493 (61.78%) | 305 (38.22%) | 0 | 0 |
| Business | 789 | 155 (19.65%) | 560 (70.99%) | 74 (9.38%) | 0 |
| Biology | 717 | 219 (30.54%) | 498 (69.46%) | 0 | 0 |
| Philosophy | 499 | 499 (100%) | 0 | 0 | 0 |
| Computer Science | 410 | 274 (66.83%) | 60 (14.63%) | 76 (18.54%) | 0 |
| History | 381 | 381 (100%) | 0 | 0 | 0 |
| Total | 12032 | 6810 (56.60%) | 4083 (33.93%) | 598 (4.97%) | 541 (4.50%) |

## A.2   5-shot CoT Prompt example

As shown as Table 7, when evaluating models on MMLU-Pro, the prompt consists of an initial prompt, 5 demonstration examples and a question to be answered. These demonstration examples are defined in the validation subset of the MMLU-Pro dataset.

Table 6: Prompt Instructions for Dataset Construction Pipeline

**STEM Website Prompt Instruction**
You are a helpful assistant to analyze a given exam question and solution. There are four things you need to do:
- **image_question**: decide whether the question requires reading a figure to answer.
- **image_answer**: decide whether the answer to the question should be a figure instead of text.
- **short_answer**: decide the question can be answered with a short phrase instead of a long sentence.
- **answer**: extract a short phrase from the solution as the answer.
- **incorrect answers**: generate three additional plausible but incorrect options as the answers, which I will use to compose a multiple-choice question.

Return your answer in JSON format like:
{ "image_question": True/False,
"image_answer": True/False,
"short_answer": True/False,
"answer": 'a short phrase as the answer',
"plausible answers": ['plausible answer 1', 'plausible answer 2', 'plausible answer 3'] }

**TheoremQA Prompt Instruction (One-shot)**
Given the following question (Q) and answer (A), please transform it into a multiple-choice question with one correct option (as provided in the answer) and three plausible distractors.
Input:
Question: What is the capital city of France?
Answer: Paris

Output:
Question: What is the capital city of France?
Options:
A) Paris
B) Madrid
C) Rome
D) Berlin

**Option Augmentation Prompt Instruction (One-shot)**
I have a multiple-choice question with four options, one of which is correct, and I need to expand it to a ten-option multiple-choice question. The original options are A, B, C, and D, with one of them being the correct answer. Please generate six additional plausible but incorrect options (E, F, G, H, I, J) to accompany the original four.
Input:
Question: What is the structure of the United Nations Security Council?
Existing 4 Options: A: 5 permanent members with veto power, 10 rotating members with no veto power
B: 5 permanent members and 10 rotating members, all with veto power
C: 10 permanent members with veto power, and 5 rotating members without veto power
D: 15 permanent members with veto power
Answer: A: 5 permanent members with veto power, 10 rotating members with no veto power

Output:
Generated 6 Options:
E: 20 members, half of whom have veto power
F: 7 permanent members with veto power, 8 rotating members with no veto power
G: 5 permanent members with no veto power, 10 rotating members with veto power
H: 15 rotating members with no veto power
I: 10 permanent members and 5 rotating members, all with veto power
J: 10 permanent members with no veto power, and 5 rotating members with veto power

**False Negative Options Recall Prompt Instruction**
You are a knowledge expert, you are supposed to answer the multi-choice question to derive your final answer as "The answer is (A)/(B)/(C)/..."

Table 7: 5-shot CoT Prompt Example in Physics

The following are multiple-choice questions (with answers) about physics. Think step by step and then finish your answer with "The answer is (X)" where X is the correct letter choice.

**Question:** A refracting telescope consists of two converging lenses separated by 100 cm. The eye-piece lens has a focal length of 20 cm. The angular magnification of the telescope is
**Options:** A. 10, B. 40, C. 6, D. 25, E. 15, F. 50, G. 30, H. 4, I. 5, J. 20
**Answer:** Let's think step by step. In a refracting telescope, if both lenses are converging, the focus of both lenses must be between the two lenses, and thus the focal lengths of the two lenses must add up to their separation. Since the focal length of one lens is 20 cm, the focal length of the other must be 80 cm. The magnification is the ratio of these two focal lengths, or 4. The answer is (H).

**Question:** Say the pupil of your eye has a diameter of 5 mm and you have a telescope with an aperture of 50 cm. How much more light can the telescope gather than your eye?
**Options:** A. 1000 times more B. 50 times more C. 5000 times more D. 500 times more E. 10000 times more F. 20000 times more G. 2000 times more H. 100 times more I. 10 times more J. N/A **Answer:** Let's think step by step. The amount of light a telescope can gather compared to the human eye is proportional to the area of its apertures. The area of a circle is given by the formula $A = \pi \left(\frac{D}{2}\right)^2$, where $D$ is the diameter. Therefore, the relative light-gathering power is calculated as:

$$\frac{\left(\frac{50 \text{ cm}}{2}\right)^2}{\left(\frac{5 \text{ mm}}{2}\right)^2} = \frac{\left(\frac{50 \text{ cm}}{0.1 \text{ cm}}\right)^2}{\left(\frac{5 \text{ mm}}{0.1 \text{ cm}}\right)^2} = \frac{500^2}{5^2} = 10000.$$

The answer is (E).

**Question:** Where do most short-period comets come from and how do we know?
**Options:** A. The Kuiper belt; short period comets tend to be in the plane of the solar system like the Kuiper belt.
B. The asteroid belt; short period comets tend to come from random directions indicating a spherical distribution of comets called the asteroid belt.
C. The asteroid belt; short period comets tend to be in the plane of the solar system just like the asteroid belt.
D. The Oort cloud; short period comets have orbital periods similar to asteroids like Vesta and are found in the plane of the solar system just like the Oort cloud.
E. The Oort Cloud; short period comets tend to come from random directions indicating a spherical distribution of comets called the Oort Cloud.
F. The Oort cloud; short period comets tend to be in the plane of the solar system just like the Oort cloud.
G. The asteroid belt; short period comets have orbital periods similar to asteroids like Vesta and are found in the plane of the solar system just like the asteroid belt.
**Answer:** Let's think step by step. Most short-period comets originate from the Kuiper belt. This is deduced from the observation that these comets tend to follow orbits that lie in the plane of the solar system, similar to the distribution of objects in the Kuiper belt itself. Thus, the alignment of these cometary orbits with the ecliptic plane points to their Kuiper belt origin. The answer is (A).

**Question:** Colors in a soap bubble result from light
**Options:** A. dispersion B. deflection C. refraction D. reflection E. interference F. converted to a different frequency G. polarization H. absorption I. diffraction J. transmission **Answer:** Let's think step by step. The colorful patterns observed in a soap bubble are caused by the phenomenon of light interference. This occurs when light waves bounce between the two surfaces of the soap film, combining constructively or destructively based on their phase differences and the varying thickness of the film. These interactions result in vibrant color patterns due to variations in the intensity of different wavelengths of light. The answer is (E).

**Question:** A microwave oven is connected to an outlet, 120 V, and draws a current of 2 amps. At what rate is energy being used by the microwave oven?
**Options:** A. 240 W B. 120 W C. 10 W D. 480 W E. 360 W F. 200 W G. 30 W H. 150 W I. 60 W J. 300 W
**Answer:** Let's think step by step. The rate of energy usage, known as power, in an electrical circuit is calculated by the product of voltage and current. For a microwave oven connected to a 120 V outlet and drawing a current of 2 amps, the power consumption can be calculated as follows:

$$\text{Power} = \text{Voltage} \times \text{Current} = 120 \text{ V} \times 2 \text{ A} = 240 \text{ W}.$$

Therefore, the microwave oven uses energy at a rate of 240 watts. The answer is (A).

**Question:** <question>
**Options:** <options>
**Answer:** Let's think step by step. <completion>

## A.3 List of Language Models Studied

In this part, we detail the model families included in our study. Our focus is on widely-used models in current production environments, such as GPT, Claude, Gemini, LLaMA, Yi [5], Phi, and other popular model families:

For closed-sourced models, we utilized the APIs of the most recent versions as of May 2024:

- **OpenAI GPT** including GPT-4o and GPT-4-Turbo. Currently strongest GPT models.
- **Anthropic Claude** including Claude-3-Opus and Claude-3-Sonnet.
- **Google Gemini** including Gemini-1.5-Pro, the most powerful model in the series, and Gemini-1.5-Flash, the newest and fastest model in the Gemini family served in the API.
- **01.AI Yi** including Yi-Large. A capable closed-source model that achieves a high score on the Chatbot Arena leaderboard.

We also examined a range of open-source base and instruction-tuned models:

- **Meta LLaMA** including Llama-3-70B-Instruct, Llama-3-70B, Llama-2-70B, Llama-3-8B-Instruct and Llama-3-8B. Important open-sourced base and instruction-tuned models.
- **Microsoft Phi** including Phi-3-medium-4k-instruct and Phi-3-mini-4k-instruct. Compact yet powerful models, excelling in knowledge and reasoning.
- **DeepSeek** including DeepSeek-V2-Chat.
- **Qwen** including Qwen1.5-110B and Qwen1.5-72B-Chat.
- **TIGER Lab MAmmoTH2** including MAmmoTH2-8x7B-Plus. A reasoning-enhanced LLM instruction tuned from Mixtral-8×7B.
- **Mistral AI Mixtral and Mistral** including Mistral-7B-v0.1 and two Mixture of Experts (MoE) models: Mixtral-8x7B-Instruct-v0.1, Mixtral-8x7B-v0.1.
- **Google Gemma** including Gemma-7B and Gemma-2B. A family of lightweight open models from Google, built from the same research and technology used to create the Gemini models.
- **01.AI Yi** including Yi-1.5-34B-Chat and Yi-34B.
- **InternLM** including InternMath-20B-Plus and InternMath-7B-Plus.
- **Other open-source LLMs** including Staring-7B, c4ai-command-r-v01, OpenChat-3.5-8B, Zephyr-7B-Beta, Neo-7B-Instruct and Llemma-7B.

## A.4 Computational Resources

Our experiments were conducted on NVIDIA A100 GPUs. To enhance the inference speed of our models, we employed the vLLM (very large language model) acceleration technique. For instance, evaluating a language model with 7 billion parameters on the MMLU-Pro dataset takes approximately 20-30 minutes. Additionally, for closed models that necessitate API calls, our evaluations on our custom dataset involved processing approximately 20M input tokens and 5M output tokens.

## A.5 Dataset Licensing

The MMLU-Pro dataset comprises data from four distinct sources, each governed by its own licensing terms:

- **MMLU dataset:** Licensed under the MIT License. This license allows for free usage, modification, and distribution, provided the original license and copyright notice are included.
- **STEM Website:** Open-Licensed.
- **TheoremQA:** Licensed under the MIT License.
- **SciBench:** Licensed under the MIT License.

Additionally, the MMLU-Pro dataset itself is licensed under the MIT License, ensuring broad usability and distribution rights under similar conditions.

---

[5] https://www.lingyiwanwu.com/

## A.6 Error Analysis Cases

In this section, we explore an error analysis of GPT-4o, currently the best-performing model on the MMLU-Pro benchmark, to examine its performance strengths and weaknesses. This examination not only highlights areas where the model falls short but also provides insights that could inform future improvements in both its architecture and training processes. We conducted a detailed review of 120 randomly selected erroneous predictions made by GPT-4o. These errors were analyzed by expert annotators who determined the underlying causes of each misprediction using their expert judgment and any definitive explanations provided.

**Reasoning Errors (39%)** Even though the model may recall correct knowledge, it often struggles to logically process steps toward the right answer. This issue often stems from logical inconsistencies in the output, possibly caused by the model's reliance on patterns from its training data rather than true understanding. For instance, as shown in Table 10, when calculating the pressure difference inside and outside a container, the model erroneously added the internal and external pressures together.

**Lack of Specific Knowledge (35%)** A fundamental root cause of domain-specific errors in the GPT-4o model is the lack of specialized knowledge. For example, as shown in Table 8, the model lacks financial knowledge: The cash balance for interest calculation is determined by subtracting the down payment from the product price. Due to the incorrect use of $1650 instead of $1600 as the principal, the result was erroneous. Similarly, as in Table 9, the model did not correctly understand that when using a lens in different media, the ratio of the refractive indices of the lens material and the medium should be considered, rather than directly subtracting their numerical values. This lack of understanding of how to properly apply optical principles led to a misconception.

**Calculation Errors (12%)** We distinguish calculation errors from reasoning errors to aid model developers, as many AI systems can utilize calculators or Python for complex, multi-step calculations. For example, as in Table 11, the model had the correct formula for calculating the molecular weight of a compound but made an error in summing the values, leading to an incorrect final answer.

**Other Errors** The remaining errors include No Selection Made (5%), Question Understanding Errors (4%), Generation Issues (2%), Annotation Errors (2%), and Answer Extraction Errors (1%). "No Selection Made" refers to instances where the model responded but did not select a final option as dictated by the prompt and few-shot format. "Question Understanding Errors" occur when the model incorrectly interprets the question or options, such as in Table 13 in the appendix, where the model incorrectly focused on Singer's broader views on equality for all beings rather than strictly on the equality principle as it applies to humans. The correct answer (E), focusing solely on human beings, was overlooked in favor of a broader interpretation (H). "Generation Issues" refer to anomalies in the generation process, such as in Table 12 in the appendix, where the model repeatedly generated one sentence until it exceeded the maximum length and terminated. "Annotation Errors" occur when the ground truth answer is incorrect. "Answer Extraction Errors" refer to failures in extracting the chosen option due to an unusual format, causing the extraction script to fail in extracting the answer choice.

## Table 8: Error Analysis Case 1: Lack of Specific Knowledge

**Question ID:** 799
**Category:** Business

**Question:**
Ms. Chen purchased a used car, worth $1650, on the installment plan, paying $50 down and $1,840 in monthly installment payments over a period of two years. What annual interest rate did she pay?
**Options:**
A. 10% B. 17.5% C. 15.2% D. 13.3% E. 20% F. 19.8% G. 18% H. 16% I. 12% J. 14.4%
**Answer:** J
**Model Predict:** C

**GPT-4o's output:**
To determine the annual interest rate Ms. Chen paid, we need to calculate the total amount paid, the principal amount, and then use the formula for the interest rate.

- Total Amount Paid:
    - Down payment: $50
    - Monthly installments: $1,840 over 2 years (24 months).
    - Total monthly payments: $1,840.
    - Total amount paid: $50 + $1,840 = $1,890.
- Principal Amount:
    - The worth of the car: $1,650.
- Interest Paid:
    - Total amount paid minus principal amount: $1,890 - $1,650 = $240.
- Monthly Payment Calculation:
    - Monthly payment: $1,840 / 24 = $76.67.
- Using the formula for the monthly interest rate:
    - Let $P$ be the principal amount ($1,650).
    - Let $r$ be the monthly interest rate.
    - Let $n$ be the number of payments (24).
    - Let $M$ be the monthly payment ($76.67).
    - The formula for the monthly payment of an installment loan is:

$$M = \frac{P \cdot r \cdot (1+r)^n}{(1+r)^n - 1}$$

- Approximate Calculation using a financial calculator:
    - Using a financial calculator or an online loan calculator, input the values:
    - Principal: $1,650.
    - Monthly payment: $76.67.
    - Number of payments: 24.
    - The monthly interest rate $r$ is approximately 1.25%.
- Annual Interest Rate:
    - Annual interest rate = Monthly interest rate $\times$ 12.
    - Annual interest rate = 1.25% $\times$ 12 = 15%.
    - Given the options, the closest to 15% is 15.2%.

**Ground Truth:**
Ms. Chen made an initial down payment of $50 and then paid $1840 in installments for the car. This results in a total payment of:

$$\$50 + \$1840 = \$1890.$$

The car had a listed cash value of $1650. Therefore, the total amount paid in installment charges is:

$$\$1890 - \$1650 = \$240.$$

However, the installment charge provided is $240. Considering the down payment, the remaining cash balance subject to installment charges was:

$$\$1650 - \$50 = \$1600.$$

To calculate the annual interest rate, we use the following formula:

$$\text{rate} = \left( \frac{2 \times \text{number of payments per year} \times \text{installment charge}}{\text{cash balance} \times (\text{number of payments to be made} + 1)} \right).$$

Substituting 12 payments per year, an installment charge of $240, a cash balance of $1600, and a total of 24 payments (12 payments per year for 2 years), we get:

$$\text{rate} = \left( \frac{2 \times 12 \times 240}{1600 \times 25} \right) = 0.144,$$

which corresponds to an annual interest rate of 14.4%.

**Explanation:**
The model lacks financial knowledge: The cash balance for interest calculation is determined by subtracting the down payment from the product price. Due to the incorrect use of $1650 instead of $1600 as the principal, the result was erroneous.

# Table 9: Error Analysis Case 2: Lack of Specific Knowledge

**Question ID:** 10277
**Category:** Physics

**Question:**
A thin lens of focal length 10 inches in air and refractive index 1.53 is immersed in water (n = 1.33). What is its focal length in water?
**Options:**
A. 20.8 inches B. 45.2 inches C. 50 inches D. 10 inches E. 15 inches F. 25.2 inches G. 18.4 inches H. 35.2 inches I. 12.6 inches J. 30 inches
**Answer:** H
**Model Predict:** F

**GPT-4o's output:**
To determine the focal length of the lens in water, we need to use the lens maker's formula, which takes into account the refractive indices of the lens material and the surrounding medium. The lens maker's formula is:

$$\frac{1}{f} = (n_{\text{lens}} - n_{\text{medium}}) \left( \frac{1}{R_1} - \frac{1}{R_2} \right)$$

Given: - Focal length in air ($f_{\text{air}}$) = 10 inches - Refractive index of the lens ($n_{\text{lens}}$) = 1.53 - Refractive index of air ($n_{\text{air}}$) = 1.00 - Refractive index of water ($n_{\text{water}}$) = 1.33
First, we find the curvature term using the focal length in air:

$$\frac{1}{f_{\text{air}}} = (n_{\text{lens}} - n_{\text{air}}) \left( \frac{1}{R_1} - \frac{1}{R_2} \right)$$

$$\frac{1}{10} = (1.53 - 1.00) \left( \frac{1}{R_1} - \frac{1}{R_2} \right)$$

$$\frac{1}{R_1} - \frac{1}{R_2} = \frac{1}{5.3}$$

Now, we use this curvature term to find the focal length in water:

$$\frac{1}{f_{\text{water}}} = (n_{\text{lens}} - n_{\text{water}}) \left( \frac{1}{R_1} - \frac{1}{R_2} \right)$$

$$\frac{1}{f_{\text{water}}} = 0.20 \left( \frac{1}{5.3} \right)$$

$$\frac{1}{f_{\text{water}}} = \frac{1}{26.5}$$

$$f_{\text{water}} = 26.5 \text{ inches}$$

However, this value does not match any of the given options. Let's re-evaluate the options and the calculations. The closest option to our calculated value is 25.2 inches. Thus, the correct answer is: The answer is (F).

**Ground Truth:**
The focal length, $f$, of a thin lens immersed in a given medium can be found by applying the equation:

$$\frac{1}{f} = (n_L - 1) \cdot \left( \frac{1}{R_1} - \frac{1}{R_2} \right)$$

where $n_L$ is the ratio of the index of the lens to the index of the medium. Since we are working with the same lens for both mediums, we can represent $\left( \frac{1}{R_1} - \frac{1}{R_2} \right)$ in equation (1) by $X$, so equation (1) becomes:

$$\frac{1}{f} = (n_L - 1)X.$$

If the focal length of the thin lens is 10 inches in air, then equation (2) becomes:

$$\frac{1}{10 \text{ in}} = \left( \frac{1.53}{1.33} - 1 \right) X$$

and if the same lens is immersed in water, equation (2) now becomes:

$$\frac{1}{f} = \left( \frac{1.53}{1.33} - 1 \right) X.$$

We now have two equations and two unknowns. To obtain the value of $f$, solve for $X$ in the first equation and substitute this value into the second. From (3) we have:

$$X = \frac{1}{10 \left( \frac{1.53}{1.33} - 1 \right)} = \frac{1}{5.3}$$

Substituting this into (4) gives us:

$$\frac{1}{f} = \left( \frac{1.53}{1.33} - 1 \right) \frac{1}{5.3};$$

hence,

$$f = 35.2 \text{ in.}$$

Thus, if a lens of focal length 10 in. in air is immersed in water, the focal length increases to about 35.2 inches.

**Explanation:**
The error made by GPT-4o in calculating the focal length of a lens in water was due to a lack of specific subject knowledge. It did not correctly understand that when using a lens in different media, the ratio of the refractive indices of the lens material and the medium should be considered, rather than directly subtracting their numerical values. This lack of understanding of how to properly apply optical principles led to a misconception.

Table 10: Error Analysis Case 3: Reasoning Errors

**Question ID:** 11826
**Category:** Engineering

**Question:**
A vertical cylinder 300 mm in diameter is fitted at the top with a tight but frictionless piston and is completely filled with water at 70°C. The outside of the piston is exposed to an atmospheric pressure of 100 kPa. Calculate the minimum force applied to the piston which will cause the water to boil.

**Options:**
A. 3.12 kN B. 7.2 kN C. 5.02 kN D. 4.00 kN E. 6.75 kN F. 1.95 kN G. 2.86 kN H. 4.86 kN I. 5.5 kN J. 3.98 kN

**Answer:** H

**Model Predict:** B

**GPT-4o's output:**
To determine the minimum force applied to the piston that will cause the water to boil at a temperature of 70°C, the following steps are taken:

- **Determine the boiling pressure at 70°C:**
  - From steam tables, the saturation pressure of water at 70°C is approximately 31.8 kPa.
- **Calculate the additional pressure needed:**
  - The atmospheric pressure is 100 kPa.
  - The total pressure required to boil the water is $31.8 \text{ kPa} + 100 \text{ kPa} = 131.8 \text{ kPa}$.
- **Calculate the area of the piston:**
  - The diameter of the piston is 300 mm, so the radius $r$ is 150 mm or 0.15 m.
  - The area $A$ of the piston is given by $A = \pi r^2$.
  - $A = \pi(0.15)^2 = \pi \times 0.0225 \approx 0.0707 \text{ m}^2$.
- **Calculate the force required:**
  - The force $F$ is given by $F = P \times A$, where $P$ is the pressure.
  - $F = 131.8 \times 10^3 \text{ Pa} \times 0.0707 \text{ m}^2$.
  - $F \approx 9317.26 \text{ N}$.
- **Convert the force to kN:**
  - $F \approx 9.317 \text{ kN}$.
- Given that the calculated force does not match any of the provided options, we re-evaluate:
  - The pressure required to boil water at 70°C is 31.8 kPa.
  - The additional pressure needed is 31.8 kPa.
  - $F = 31.8 \times 10^3 \text{ Pa} \times 0.0707 \text{ m}^2 \approx 2247.66 \text{ N} \approx 2.25 \text{ kN}$.
- However, upon rechecking the options, the closest value is:
  - **The answer is (B) 7.2 kN** given the confusion in the problem statement.

**Ground Truth:**
The setup involves a piston that must be withdrawn slowly from a cylinder to avoid acceleration. Since the water in the cylinder cannot expand, a space filled with water vapor will be created beneath the piston, causing the water to boil. The vapor pressure of water at 70°C is critical to this process. Given:
- Vapor Pressure of water at 70°C: 31.2 kPa. - Atmospheric pressure outside the piston: 100 kPa. - Diameter of the piston: 300 mm (or 0.3 m).
The pressure difference across the piston is the atmospheric pressure minus the vapor pressure inside:

$$P = 100 \text{ kPa} - 31.2 \text{ kPa} = 68.8 \text{ kPa}$$

The area over which this pressure acts is calculated based on the diameter of the piston:

$$A = \frac{\pi(0.3)^2}{4} = \frac{\pi \times 0.09}{4} = \frac{0.0225\pi}{1} \approx 0.0706858 \text{ m}^2$$

Using the formula $P = \frac{F}{A}$, we solve for the force $F$:

$$F = P \times A = 68.8 \times 10^3 \text{ Pa} \times 0.0706858 \text{ m}^2 = 4864.8 \text{ N}$$

Converting to kilonewtons (kN):

$$F \approx 4.8648 \text{ kN}$$

Thus, the force that must be applied to the piston, to maintain the conditions for boiling at 70°C, is approximately 4.86 kN.

**Explanation:**
The model has a reasoning error; when calculating the pressure difference, the pressure inside and outside the container should be subtracted, not added.

## Table 11: Error Analysis Case 4: Calculation Errors

**Question ID:** 4138
**Category:** Chemistry

**Question:**
The gypsy moth produces a natural attractant, $C_{18}H_{34}O_3$. If a female moth is trapped behind a cellophane screen containing a pinhole and the carbon dioxide she produces diffuses through the pinhole at the rate of 1 millimicromole per 90 seconds, what quantity of attractant will diffuse through the orifice in the same amount of time (90 seconds)?

**Options:**
A. 0.25 millimicromoles B. 0.45 millimicromoles C. 0.40 millimicromoles D. 0.48 millimicromoles E. 0.42 millimicromoles
F. 0.30 millimicromoles G. 0.35 millimicromoles H. 0.34 millimicromoles I. 0.38 millimicromoles J. 0.50 millimicromoles

**Answer:** I

**Model Predict:** B

**GPT-4o's output:**
To solve this problem, we need to apply Graham's law of effusion, which states that the rate of effusion of a gas is inversely proportional to the square root of its molar mass. The formula for Graham's law can be written as:

$$\frac{r_1}{r_2} = \sqrt{\frac{M_2}{M_1}}$$

where $r_1$ and $r_2$ are the effusion rates of gases 1 and 2, respectively, and $M_1$ and $M_2$ are their molar masses.

Given: - The effusion rate of CO2 ($r_{CO2}$) is 1 millimicromole per 90 seconds. - The molar mass of CO2 ($M_{CO2}$) is 44 g/mol. - The molar mass of the attractant ($M_{attractant}$) is calculated as $18 \times 12 + 34 \times 1 + 3 \times 16 = 218$ g/mol.

We need to calculate the effusion rate of the attractant ($r_{attractant}$).

Applying Graham's law:

$$\frac{r_{CO2}}{r_{attractant}} = \sqrt{\frac{M_{attractant}}{M_{CO2}}}$$

Calculate the square root:

$$\sqrt{\frac{218}{44}} \approx \sqrt{4.9545} \approx 2.225$$

So,

$$\frac{1 \text{ millimicromole per 90 seconds}}{r_{attractant}} = 2.225$$

Therefore,

$$r_{attractant} = \frac{1 \text{ millimicromole per 90 seconds}}{2.225} \approx 0.449 \text{ millimicromoles per 90 seconds}$$

Rounding to two decimal places, we get:

$$r_{attractant} \approx 0.45 \text{ millimicromoles per 90 seconds}$$

**Ground Truth:**
The rate of escape of gases through an orifice is inversely proportional to the square root of the density of the gas. This principle is known as Graham's Law. If two gases are compared at the same temperature and pressure (as in this problem), then

$$\frac{u_1}{u_2} = \sqrt{\frac{d_2}{d_1}}$$

where $u$ is the velocity (or the rate at which the gas diffuses out of the orifice) and $d$ is the density of the particular gas. To solve this problem, this equation is modified somewhat to avoid using densities in the expression (since the densities are unknown). However, the density is defined as mass per unit volume, and since the volume per mole of both gases, $C_{18}H_{34}O_3$ and $CO_2$, are the same, Graham's Law becomes

$$\frac{u_1}{u_2} = \sqrt{\frac{d_2}{d_1}} = \sqrt{\left(\frac{m_2/V}{m_1/V}\right)} = \sqrt{\frac{m_2}{m_1}},$$

where $V$ is volume and $m_1$ and $m_2$ are the masses of the gases.

Another modification is necessary before solving this problem: the mass of any substance is its molecular weight times the number of moles of that substance. However, assume that 1 mole is involved in both substances and thus molecular weights of the gases (which can be determined) are substituted. Graham's Law then becomes

$$\frac{u_1}{u_2} = \sqrt{\frac{M_2}{M_1}}$$

$u_1$ for $CO_2 = 10^{-9}$ mole/90 sec $= 1.1 \times 10^{-11}$ moles/sec.
Note: 1 milli micromole $= 10^{-9}$ mole.
$M_1$ for $CO_2 = 44$ g/mole
$M_2$ for $C_{18}H_{34}O_3 = 298$ g/mole.
Thus, solve for $u_2$:

$$\frac{1.1 \times 10^{-11} \text{ moles/sec}}{u_2} = \sqrt{\left(\frac{298 \text{ g/mole}}{44 \text{ g/mole}}\right)} = 2.6$$

$$u_2 = 4.2 \times 10^{-12} \text{ moles/sec}$$

1 picomole $= 10^{-12}$ moles.
$u_2 = 4.2 \times 10^{-12}$ moles/sec $= 4.2$ picomoles per sec
$u_1 = 1.1 \times 10^{-11}$ moles/sec $= 11.0$ picomoles per sec.
The $CO_2$ diffuses 1 millimicromole within 90 sec $[0.011$ millimicromole/sec$][90$ sec$] = 1$ millimicromole and $C_{18}H_{34}O_3$ diffuses $(0.0042$ millimicromoles/sec$)(90$ sec$) = 0.38$ millimicromoles within the same time as the $CO_2$.

**Explanation:**
The molecular weight calculation error in GPT-4o's output is incorrect. The correct molecular weights should be calculated as 12 * 18 + 1 * 34 + 16 * 3 = 298 g/mol instead of 218 g/mol in the GPT-4o's output.

**Question ID:** 4138
**Category:** Chemistry

**Question:**
There are two highly inbred strains of laboratory mice whose adult body weights are very different. Assume that the mouse's body weight is under the control of three pairs of contrasting genes: $A$ vs. $a$, $B$ vs. $b$, and $D$ vs. $d$. Assume further that each capital letter gene is responsible for contributing 5.0 grams to the total body weight, and that lowercase letters contribute 2.5 grams to the total body weight. The average weight of mice in Strain I is 30 grams, while that of Strain II mice is 15 grams.
(a) What are the most likely genotypes of these two strains?
(b) Suppose Strain I and Strain II are crossed. What will be the phenotype of the $F_1$ progeny?
**Options:**
A. Strain I: $AABBDD$, Strain II: $aabbdd$, $F_1$ progeny: 20g
B. Strain I: $AAbbDD$, Strain II: $aaBBdd$, $F_1$ progeny: 25g
C. Strain I: $AABBDD$, Strain II: $aabbdd$, $F_1$ progeny: 22.5g
D. Strain I: $AABB$, Strain II: $aabb$, $F_1$ progeny: 20g
E. Strain I: $AABB\,Dd$, Strain II: $aabb\,DD$, $F_1$ progeny: 21.25g
F. Strain I: $AABBDD$, Strain II: $aaBBdd$, $F_1$ progeny: 30g
G. Strain I: $AABBDD$, Strain II: $aabbDD$, $F_1$ progeny: 27.5g
H. Strain I: $Aa\ Bb\ Dd$, Strain II: $Aa\ Bb\ Dd$, $F_1$ progeny: 30g
I. Strain I: $AAbb\ dd$, Strain II: $aaBB\ DD$, $F_1$ progeny: 22.5g
J. Strain I: $AABB\ DD$, Strain II: $aabb\ dd$, $F_1$ progeny: 25g
**Answer:** C
**Model Predict:** Null

**GPT-4o's Output:**
Let's break down the problem step by step:
1. **Determine the genotypes of Strain I and Strain II:** - Strain I has an average weight of 30 grams. - Strain II has an average weight of 15 grams.
2. **Calculate the contribution of each gene to the body weight:** - Each capital letter gene (A, B, D) contributes 5.0 grams. - Each lowercase letter gene (a, b, d) contributes 2.5 grams.
3. **Determine the total genetic contribution to the body weight:** - For Strain I (30 grams): - $30 = (A + A + B + B + D + D)$ - Each capital letter contributes 5 grams, so: - $30 = 6 \times 5$ - Therefore, Strain I must have the genotype $AABBDD$.
- For Strain II (15 grams): - $15 = (a + a + b + b + d + d)$ - Each lowercase letter contributes 2.5 grams, so: - $15 = 6 \times 2.5$ - Therefore, Strain II must have the genotype $aabbdd$.
4. **Determine the phenotype of the F1 progeny:** - When Strain I (AABB DD) is crossed with Strain II (aabb dd), the F1 progeny will have the genotype $AaBbDd$. - Each heterozygous pair (Aa, Bb, Dd) will contribute the average of the contributions of the dominant and recessive alleles: - $A = 5, a = 2.5 \rightarrow Aa = (5 + 2.5)/2 = 3.75$ - $B = 5, b = 2.5 \rightarrow Bb = (5 + 2.5)/2 = 3.75$ - $D = 5, d = 2.5 \rightarrow Dd = (5 + 2.5)/2 = 3.75$
- Total weight contribution from each heterozygous pair: - $3.75 + 3.75 + 3.75 = 11.25$ grams per pair. - Since there are three pairs, the total weight is $11.25 \times 3 = 33.75$ grams.
5. **Recalculate the phenotype of the F1 progeny:** - The correct calculation should be: - $Aa = 3.75, Bb = 3.75, Dd = 3.75$ - Total weight contribution: $3.75 \times 3 = 11.25$ grams per pair. - Since there are three pairs, the total weight is $11.25 \times 3 = 33.75$ grams.
- However, this seems incorrect. Let's re-evaluate: - The correct calculation should be: - $Aa = 3.75, Bb = 3.75, Dd = 3.75$ - Total weight contribution: $3.75 \times 3 = 11.25$ grams per pair. - Since there are three pairs, the total weight is $11.25 \times 3 = 33.75$ grams.
- The correct calculation should be: - $Aa = 3.75, Bb = 3.75, Dd = 3.75$ - Total weight contribution: $3.75 \times 3 = 11.25$ grams per pair. - Since there are three pairs, the total weight is $11.25 \times 3 = 33.75$ grams.
- The correct calculation should be: - $Aa = 3.75, Bb = 3.75, Dd = 3.75$ - Total weight contribution: $3.75 \times 3 = 11.25$ grams per pair. - Since there are three pairs, the total weight is $11.25 \times 3 = 33.75$ grams.
- The correct calculation should be: - $Aa = 3.75, Bb = 3.75, Dd = 3.75$ - Total weight contribution: $3.75 \times 3 = 11.25$ grams per pair. - Since there are three pairs, the total weight is $11.25 \times 3 = 33.75$ grams.
- The correct calculation should be: - $Aa = 3.75, Bb = 3.75, Dd = 3.75$ - Total weight contribution: $3.75 \times 3 = 11.25$ grams per pair. - Since there are three pairs, the total weight is $11.25 \times 3 = 33.75$ grams.
- The correct calculation should be: - $Aa = 3.75, Bb = 3.75, Dd = 3.75$ - Total weight contribution: $3.75 \times 3 = 11.25$ grams per pair. - Since there are three pairs, the total weight is $11.25 \times 3 = 33.75$ grams.
- The correct calculation should be: - $Aa = 3.75, Bb = 3.75, Dd = 3.75$ - Total weight contribution: $3.75 \times 3 = 11.25$ grams per pair. - Since there are three pairs, the total weight is $11.25 \times 3 = 33.75$ grams.
- The correct calculation should be: - $Aa = 3.75, Bb = 3.75, Dd = 3.75$ - Total weight contribution: $3.75 \times 3 = 11.25$ grams per pair. - Since there are three pairs, the total weight is $11.25 \times 3 = 33.75$ grams.
- The correct calculation should be: - $Aa = 3.75, Bb = 3.75, Dd = 3.75$ - Total weight contribution: $3.75 \times 3 = 11.25$ grams per pair. - Since there are three pairs, the total weight is $11.25 \times 3 = 33.75$ grams.
- The correct calculation should be: - $Aa = 3.75, Bb = 3.75, Dd = 3.75$ - Total weight contribution: $3.75 \times 3 = 11.25$ grams per pair. - Since there are three pairs, the total weight is $11.25 \times 3 = 33.75$ grams.
- The correct calculation should be: - $Aa = 3.75, Bb = 3.75, Dd = 3.75$ - Total weight contribution: $3.75 \times 3 = 11.25$ grams per pair. - Since there are three pairs, the total weight is $11.25 \times 3 = 33.75$ grams.
- The correct calculation should be: - $Aa = 3.75, Bb = 3.75, Dd = 3.75$ - Total weight contribution: $3.75 \times 3 = 11.25$ grams per pair. - Since there are three pairs, the total weight is $11.25 \times 3 = 33.75$ grams.
- The correct calculation should be: - $Aa = 3.75, Bb = 3.75, Dd = 3.75$ - Total weight contribution: $3.75 \times 3 = 11.25$ grams per pair. - Since there are three pairs, the total weight is $11.25 \times 3 = 33.75$ grams.
- The correct calculation should be: - $Aa = 3.75, Bb = 3.75, Dd = 3.75$ - Total weight contribution: $3.75 \times 3 = 11.25$ grams per pair. - Since there are three pairs, the total weight is $11.25 \times 3 = 33.75$ grams.
- The correct calculation should be: - $Aa = 3.75, Bb = 3.75, Dd = 3.75$ - Total weight contribution: $3.75 \times 3 = 11.25$ grams per pair. - Since there are three pairs, the total weight is $11.25 \times 3 = 33.75$ grams.
- The correct calculation should be: - $Aa = 3.75, Bb = 3.75, Dd = 3.75$ - Total weight contribution: $3.75 \times 3 = 11.25$ grams per pair. - Since there are three pairs, the total weight is $11.25 \times 3 = 33.75$ grams.
- The correct calculation should be: - $Aa = 3.75, Bb = 3.75, Dd = 3.75$ - Total weight contribution: $3.75 \times 3 = 11.25$ grams per pair. - Since there are three pairs, the total weight

**Explanation:**
The model repeatedly regenerates a phrase during the generation process until the generation is complete.

Table 13: Error Analysis Case 6: Question Understanding Errors

**Question ID:** 11236
**Category:** Philosophy

**Question:**
In Singer's understanding, the principle of the equality of human beings is
**Options:**
A. a description of an alleged actual equality among humans.
B. a description of an alleged equality among all living beings.
C. a prescription of how we should treat nonhuman animals.
D. a description of an alleged inequality among all living beings.
E. a prescription of how we should treat humans.
F. a description of an alleged actual inequality among humans.
G. a description of an alleged actual superiority of humans over nonhuman animals.
H. a prescription of how we should treat both human and nonhuman animals equally.
I. a prescription of how we should treat nonhuman animals differently.
J. a prescription of how we should treat the environment.
**Answer:** E
**Model Predict:** H

**GPT-4o's output:**
To determine the correct answer, we need to understand Peter Singer's principle of the equality of human beings. Singer is a well-known philosopher, particularly in the field of ethics, and he has written extensively on the topic of equality and animal rights.

Singer's principle of equality is not about asserting that all humans are actually equal in abilities or characteristics. Instead, it is a moral principle prescribing how we should treat beings. Specifically, Singer argues that we should give equal consideration to the interests of all beings capable of suffering, whether they are human or nonhuman animals.

Given this understanding, let's evaluate the options:

A. a description of an alleged actual equality among humans. - This is incorrect because Singer's principle is not about describing actual equality among humans.
B. a description of an alleged equality among all living beings. - This is incorrect because Singer's principle is not about describing equality among all living beings.
C. prescription of how we should treat nonhuman animals. - This is partially correct but not comprehensive enough.
D. a description of an alleged inequality among all living beings. - This is incorrect because Singer's principle is not about describing inequality.
E. a prescription of how we should treat humans. - This is partially correct but not comprehensive enough.
F. a description of an alleged actual inequality among humans. - This is incorrect because Singer's principle is not about describing actual inequality.
G. a description of an alleged actual superiority of humans over nonhuman animals. - This is incorrect because Singer argues against the superiority of humans over nonhuman animals.
H. a prescription of how we should treat both human and nonhuman animals equally. - This is correct because it captures the essence of Singer's principle, which is about giving equal consideration to the interests of all beings capable of suffering.
I. a prescription of how we should treat nonhuman animals differently. - This is incorrect because Singer advocates for equal consideration, not different treatment.
J. a prescription of how we should treat the environment. - This is incorrect because Singer's principle specifically addresses the treatment of beings capable of suffering, not the environment in general.
The best answer is: The answer is H.

**Explanation:**
The model incorrectly interpreted the question, focusing on Singer's broader views on equality for all beings rather than strictly on the equality principle as it applies to humans. The correct answer (E), focusing solely on human beings, was overlooked in favor of a broader interpretation (H).

