# Supplementary Material of MMLU-Pro Benchmark

[1]Yubo Wang*, [1]Xueguang Ma*, [1]Ge Zhang, [1]Yuansheng Ni, [1]Abhranil Chandra,
[1]Shiguang Guo, [1]Weiming Ren, [1]Aaran Arulraj, [1]Xuan He, [1]Ziyan Jiang, [1]Tianle Li,
[1]Max Ku, [2]Kai Wang, [1]Alex Zhuang, [1]Rongqi Fan, [3]Xiang Yue, [1]Wenhu Chen*,

[1]University of Waterloo, [2]University of Toronto, [3]Carnegie Mellon University,

## 1 Dataset Accessibility

- URL to website/platform where the dataset/benchmark can be viewed and downloaded by the reviewers:
  https://huggingface.co/datasets/TIGER-Lab/MMLU-Pro

- URL to Croissant metadata record documenting the dataset/benchmark available for viewing and downloading by the reviewers:
  https://huggingface.co/api/datasets/TIGER-Lab/MMLU-Pro/croissant

- To reproduce our experimental results, see our GitHub repository at:
  https://github.com/TIGER-AI-Lab/MMLU-Pro

- The persistent dereferenceable identifier (DOI) of our dataset:
  10.57967/hf/2439

## 2 Datasheets for Datasets

To illustrate the Dataset documentation and intended uses, we have completed the following fill-in of the Datasheets for Datasets:

### 2.1 Motivation

- **For what purpose was the dataset created?**

  To provide a more challenging and robust benchmark for multi-task language understanding evaluation.

- **Who created the dataset (e.g., which team, research group) and on behalf of which entity (e.g., company, institution, organization)?**

  TIGER Lab, University of Waterloo

- **Who funded the creation of the dataset?**

  University of Waterloo

### 2.2 Composition

- **What do the instances that comprise the dataset represent (e.g., documents, photos, people, countries)?**

  The dataset consists of multiple-choice questions from various subjects, each instance representing a question, its answer options, and metadata such as question source and subject category.

- **How many instances are there in total (of each type, if appropriate)?**

  12032 instances.

---

*Core Contributors. ✉: {y726wang, x93ma, wenhuchen}@uwaterloo.ca

38th Conference on Neural Information Processing Systems (NeurIPS 2024) Track on Datasets and Benchmarks.

- **Does the dataset contain all possible instances or is it a sample (not necessarily random) of instances from a larger set?**

  All possible instances.

- **What data does each instance consist of?**

  Each instance consists of a question, options, its answer, and metadata such as question source and subject category.

- **Is there a label or target associated with each instance?**

  Yes, each question instance in the dataset has an associated label, which is the correct answer to the question.

- **Is any information missing from individual instances?**

  No.

- **Are relationships between individual instances made explicit (e.g., users' movie ratings, social network links)?**

  No, the individual instances in this dataset do not have explicit relationships between them. Each instance is treated independently without direct connections to others, suitable for analyses or models that do not require relational data.

- **Are there recommended data splits (e.g., training, development/validation, testing)?**

  Yes, the dataset includes recommended splits for testing and validation purposes.

- **Are there any errors, sources of noise, or redundancies in the dataset?**

  Yes, there may be errors and sources of noise in the dataset, particularly because it includes manually annotated data.

- **Is the dataset self-contained, or does it link to or otherwise rely on external resources (e.g., websites, tweets, other datasets)?**

  Yes, it is self-contained.

- **Does the dataset contain data that might be considered confidential (e.g., data that is protected by legal privilege or by doctor-patient confidentiality, data that includes the content of individuals' non-public communications)?**

  No.

- **Does the dataset contain data that, if viewed directly, might be offensive, insulting, threatening, or might otherwise cause anxiety?**

  No.

## 2.3 Collection Process

- **How was the data associated with each instance acquired?**

  The data associated with each instance was indirectly inferred or derived from other data. To ensure the accuracy and reliability of this inferred data, it was validated and verified through manual annotation. This process involved human annotators reviewing the derived data to correct any inaccuracies and confirm its validity.

- **What mechanisms or procedures were used to collect the data (e.g., hardware apparatuses or sensors, manual human curation, software programs, software APIs)?**

  The data was collected using a combination of software APIs and manual human curation. We utilized several third-party APIs to automatically fetch data from various online platforms, which was then processed and structured by our software tools. To ensure the data's accuracy and relevance, a team of human curators reviewed and annotated the data. They corrected any discrepancies and enriched the dataset with additional metadata.

- **If the dataset is a sample from a larger set, what was the sampling strategy (e.g., deterministic, probabilistic with specific sampling probabilities)?**

  No, it is not a sample from a larger set.

- **Who was involved in the data collection process (e.g., students, crowd workers, contractors) and how were they compensated (e.g., how much were crowd workers paid)?**
  All the coauthors of the article were involved in the data collection process for our study. As members of the academic team, they contributed to various aspects of the data collection. The co-authors were not compensated financially for their roles in data collection; their involvement was part of their research duties and contributions toward the publication of the study.

- **Over what timeframe was the data collected?**
  The data collection period for our study spanned from March 2024 to April 2024.

- **Were any ethical review processes conducted (e.g., by an institutional review board)?**
  N/A

## 2.4 Preprocessing/cleaning/labeling

- **Was any preprocessing/cleaning/labeling of the data done (e.g., discretization or bucketing, tokenization, part-of-speech tagging, SIFT feature extraction, removal of instances, processing of missing values)?**
  The data underwent a manual annotation process.

- **Was the "raw" data saved in addition to the preprocessed/cleaned/labeled data (e.g., to support unanticipated future uses)?**
  Yes.

- **Is the software that was used to preprocess/clean/label the data available?**
  Yes.

## 2.5 Uses

- **Has the dataset been used for any tasks already?**
  Yes, the dataset has already been used to evaluate numerous language models.

- **Is there a repository that links to any or all papers or systems that use the dataset?**
  Yes, https://huggingface.co/datasets/TIGER-Lab/MMLU-Pro.

- **What (other) tasks could the dataset be used for?**
  Currently, there are no other identified uses for the dataset.

- **Is there anything about the composition of the dataset or the way it was collected and preprocessed/cleaned/labeled that might impact future uses?**
  We have conducted thorough manual checks on the dataset. Any minor errors that might exist will have minimal impact on future uses.

- **Are there tasks for which the dataset should not be used?**
  Our dataset should only be used for evaluating language models.

## 2.6 Distribution

- **Will the dataset be distributed to third parties outside of the entity (e.g., company, institution, organization) on behalf of which the dataset was created?**
  Yes.

- **How will the dataset will be distributed (e.g., tarball on website, API, GitHub)?**
  GitHub and Huggingface.
  https://github.com/TIGER-AI-Lab/MMLU-Pro    https://huggingface.co/datasets/TIGER-Lab/MMLU-Pro

- **When will the dataset be distributed?**
  The dataset has already been distributed.

- **Will the dataset be distributed under a copyright or other intellectual property (IP) license, and/or under applicable terms of use (ToU)?**

The dataset will be distributed without any copyright restrictions and will be open source. It is permissible for anyone to use it for research, study, or other non-commercial purposes.

- **Have any third parties imposed IP-based or other restrictions on the data associated with the instances?**

  No.

- **Do any export controls or other regulatory restrictions apply to the dataset or to individual instances?**

  No.

### 2.7 Maintenance

- **Who will be supporting/hosting/maintaining the dataset?**

  TIGER Lab, University of Waterloo

- **How can the owner/curator/manager of the dataset be contacted (e.g., email address)?**

  y726wang@uwaterloo.ca, x93ma@uwaterloo.ca, wenhuchen@uwaterloo.ca

- **Is there an erratum?**

  We do not have an errata, but we will continuously update the content on the Hugging Face platform. The corrections for any errors or omissions can be found on Hugging Face.

- **Will the dataset be updated (e.g., to correct labeling errors, add new instances, delete instances)?**

  Yes, we will continuously update on the Hugging Face platform.

- **If the dataset relates to people, are there applicable limits on the retention of the data associated with the instances (e.g., were the individuals in question told that their data would be retained for a fixed period of time and then deleted)?**

  The dataset does not involve any personal information.

- **Will older versions of the dataset continue to be supported/hosted/maintained?**

  In fact, our dataset will not undergo major version updates.

- **If others want to extend/augment/build on/contribute to the dataset, is there a mechanism for them to do so?**

  To extend, augment, or build on the dataset, others simply need to cite our work. For contributions, please contact us on Hugging Face or GitHub.