# OpenReview forum: "MMLU-Pro: A More Robust and Challenging Multi-Task Language Understanding Benchmark"
_NeurIPS.cc/2024/Datasets_and_Benchmarks_Track — NeurIPS 2024 Track Datasets and Benchmarks Spotlight_

### Official Review · Reviewer_6sZ1 · 2024-07-12
**Review for Paper #913**

**Rating:** 6
**Confidence:** 4
**Correctness:** Yes!
**Clarity:** Yes!

**Review:**

The authors present a thorough and methodically sound approach to enhancing the MMLU benchmark. The process of dataset construction is detailed, including initial filtering, question collection and integration, option augmentation, and expert review. The inclusion of extensive experimental results with various LLMs further solidifies the credibility and reliability of the dataset. While building upon the existing MMLU benchmark, the authors introduce significant enhancements that address its limitations. The expansion of answer choices and the emphasis on reasoning-focused questions represent innovative steps toward creating a more challenging and discriminative benchmark.

**Strengths:**

1. The proposed dataset introduces more challenging questions and expands answer choices, reducing the likelihood of guessing.

2. They perform rigorous quality control by implementing two rounds of expert reviews to eliminate trivial, noisy, or incorrect questions.

3. One of the key bottlenecks in LLM research is sensitivity to prompt changes and the authors demonstrate reduced sensitivity to prompt variations, ensuring stable performance measurements.

4. The authors emphasize on reasoning over knowledge recall, i.e., requiring models to perform deliberate reasoning to answer correctly.

**Additional Feedback:**

N/A

**Documentation:**

Yes!

**Limitations:**

It would be great if the authors could comment on the following open questions:

1. There could be an implicit data source bias, i.e., the questions might still reflect inherent biases from the sources used, despite rigorous filtering and review.

2. The dataset may involve a degree of subjectivity from the experts, which might influence the final dataset.

**Weaknesses:**

1. More examples of the newly added challenging questions and reasoning-focused questions to contrast with the existing MMLU dataset could enhance the understanding and significance of the proposed dataset.

2. While the enhancements to the MMLU benchmark are significant, they might be viewed as incremental improvements rather than a completely novel approach. Moreover, the originality is primarily in question complexity and choice expansion; it does not introduce entirely new types of tasks or evaluation methods.

3. The benchmark's increased difficulty might render it less useful for evaluating lower-performing or smaller-scale models, thus limiting its applicability.

4. There's a risk that models might overfit to the specific types of reasoning required by MMLU-Pro, potentially neglecting other important capabilities in language understanding.

**Opportunities For Improvement:**

Please see the limitations for more details.

**Relation To Prior Work:**

Yes!

**Summary And Contributions:**

The paper introduces MMLU-Pro, an enhanced dataset aimed at evaluating advanced Large Language Models (LLMs). MMLU-Pro addresses the limitations of the original MMLU benchmark by incorporating more challenging, reasoning-focused questions and expanding the answer choices from four to ten. This new dataset reduces trivial questions, adds complex college-level problems, and undergoes rigorous expert review to ensure quality. Experimental results demonstrate that MMLU-Pro increases difficulty, decreases accuracy by 16-33%, and reduces sensitivity to prompt variations, making it a more discriminative and robust benchmark for evaluating LLMs.

---

> ### Author Rebuttal · Authors · 2024-08-15
>
> Thank you for your comprehensive review and thoughtful insights on our MMLU-Pro paper. We appreciate your recognition of the paper's strengths and would like to address your concerns and suggestions:
>
> 1. Examples of challenging questions:
> We acknowledge this oversight and agree that providing more examples would enhance the understanding of our dataset. In the revised manuscript, we will include a comparative section with examples of new challenging questions alongside original MMLU questions to better illustrate the improvements.
>
> 2. Incremental vs. novel approach:
> While we understand your perspective, we'd like to emphasize that our approach led to significant scientific discoveries in LLM evaluation. By introducing more challenging questions and expanding answer choices from four to ten, we observed a substantial impact on model performance. This caused a significant drop in accuracy by 16% to 33% compared to MMLU, demonstrating the increased difficulty of MMLU-Pro. Moreover, our option augmentation significantly improved the benchmark's robustness. In our experiments with 24 different prompt styles, we found that the sensitivity of model scores to prompt variations was just 2% in MMLU-Pro. This is a notable improvement in stability compared to existing benchmarks.
>
> These findings represent valid scientific discoveries that fundamentally alter the nature of LLM evaluation, making MMLU-Pro substantially more challenging, discriminative, and robust. We believe these advancements address critical limitations in existing benchmarks and provide valuable insights into model capabilities and areas for improvement.
>
> 3. Applicability to smaller models:
> We appreciate this concern. However, our leaderboard includes various smaller models that show good differentiation, such as Gemma-2B (15.85% accuracy), Qwen2-1.5B (22.56% accuracy), and Phi3-mini-4k (45.66% accuracy). This demonstrates that MMLU-Pro maintains its discriminative power across a wide range of model sizes. We'll include a more detailed discussion of this in our revised paper.
>
> 4. Risk of overfitting to specific reasoning types:
> The evaluation of LLMs typically involves a wide range of test sets to demonstrate their capabilities across various aspects. For instance, the Hugging Face Open LLM Leaderboard adopts multiple evaluation sets such as IFEval, BBH, MMLU-Pro, MATH, and GPQA to provide a more comprehensive assessment of model performance.
>
> Regarding the limitations you mentioned:
>
> 1. Data source bias: We are exploring other sources to diversify our question bank in future versions of MMLU-Pro. Additionally, we are committed to the long-term maintenance of MMLU-Pro's question set, continuously working to correct any biases within it.
>
> 2. Expert subjectivity: As for the concern about potential expert subjectivity, we would like to clarify several points: Firstly, all questions in MMLU-Pro are multiple-choice, not open-ended, which inherently reduces subjectivity in evaluation. Secondly, for STEM subjects, which comprise a significant portion of our dataset, the answers are objective and verifiable. In humanities and social sciences, where questions might involve more nuance, our experts followed a rigorous process. When faced with uncertainty, they consulted relevant literature and engaged in collaborative discussions to reach a consensus.
>
> Thank you again for your insightful feedback. We believe addressing these points will significantly strengthen our paper.

---

> > ### Comment · Reviewer_6sZ1 · 2024-08-24
> >
> > Thank you for addressing my feedback! Considering the rating from other reviewers, I would like to retain my score.

---

### Official Review · Reviewer_WZvx · 2024-07-26

**Rating:** 7
**Confidence:** 3
**Clarity:** The paper presentation is good.

**Review:**

**Pros:**
* The issues with the existing benchmarks are clearly explained, which provide good motivations of the importance of MMLU-Pro.
* The differences of MMLU-Pro vs. MMLU are explained in detail.
* While MMLU-Pro is of higher scale compared to MMLU, extensive experiments have also been conducted to investigate its efficiency. The experiments were run on 50 open- or closed-source LLMs.
* The key findings of the conducted comprehensive benchmarking are clearly and well presented.
* The experiments are investigated and the results are interpreted in a meaningful manner.

**Cons:**
* How is it possible to extend the data sources to include other and more diverse source of the data? Currently, in addition to the Original MMLU Questions, the majority of data comes from STEM Website. Is it possible to extend this to other related websites?
* It could be valuable to compare the size and other important characteristics of MMLU vs. MMLU-Pro in a table.
* There is a filtering strategy for removing the “too easy” question from the original MMLU questions. Why the same approach has not been applied to the questions coming from STEM Website, TheoremQA, and SciBench.
* It could be insightful to extend the experiments, e.g., what is the impact of the number of shots on the model performance?
* The privacy issues as well as potential negative social impacts are not discussed.

**Strengths:**

The paper is well presented. The contributions are well motivated, showing the importance of the work. The dataset is larger and more complex than the existing benchmarks. There are extensive experiments on the contributed data and the performance of more than 50 LLMs are examined on this benchmark.

While data comes from previous work or public resources, the ethical, privacy, or social implications are not discussed.

**Additional Feedback:**

There are som changes pointed out in the **Review** section.

**Correctness:**

The paper is well-written and easy to follow. It also explains the data generation and benchmarking in a clear way. All procedures are conducted in a sound way.
Also, the experiments aligns with the goal of the paper and provides a very interesting comparison among most of the current successful models.

**Documentation:**

The details of the dataset construction, organization, and experimental analysis are sufficient.
The maintenance and availability plan is not discussed.
The important and applicability of the MMLU-Pro in the realm of LLMs and its relation to ethical and responsible AI are not discussed.

**Ethics:**

The data is scrapped from public resources or existing datasets.
It does not see to be an ethical issue with the work, although it is not discussed.

**Limitations:**

Some of the design choices are not well explained. For example, it is not clear what alternative for STEM Websites were available, and why the current setting has been adopted.

**Opportunities For Improvement:**

Some discussed on the social negative impacts of the datasets should be added.

**Relation To Prior Work:**

The prior work and their issues are discussed. Why this work is important and how it is different from previous works are well motivated.

**Summary And Contributions:**

The Massive Multitask Language Understanding (MMLU) benchmark has a fundamental role in the performance improvement of LLMs. This paper presents MMLU-Pro which extends the MMLU benchmark by incorporating more challenging, reasoning-focused questions as well as expanding the choice set. MMLU-Pro also eliminates the trivial and noise questions in MMLU. By raising the challenges, MMLU-Pro causes drop in performance as well as sensitivity of the model scores to prompt variations. It is shown that Chain-of-Thought (CoT) achieves higher performance than direct answering on MMLU-Pro which indicates the more complex reasoning of MMLU-Pro questions.

---

> ### Author Rebuttal · Authors · 2024-08-15
>
> Thank you for your thorough review and positive feedback on our MMLU-Pro paper. We appreciate your recognition of the paper's strengths and would like to address your concerns and suggestions:
>
> 1. Extending data sources:
> We agree that diversifying our data sources could enhance the benchmark. While we focused on STEM websites to ensure high-quality, specialized content, we are exploring the incorporation of additional reputable sources in future iterations of MMLU-Pro to broaden its scope and representation.
> 2. Comparison table:
> Thank you for this suggestion. We will add a comprehensive comparison table of MMLU vs. MMLU-Pro characteristics in the revised manuscript to provide a clear overview of the differences.
>
> 3. Filtering strategy for non-MMLU sources:
> The newly added questions from the STEM Website, TheoremQA, and SciBench are predominantly college-level questions, which generally reduces the occurrence of "too easy" questions. Additionally, maintaining some simpler questions is crucial for differentiating between small-scale or lower-performing models. However, we acknowledge that applying a consistent filtering strategy across all sources could be beneficial and will consider implementing this in future versions of the dataset.
>
> 4. Impact of shot number:
> We appreciate this insightful suggestion. In response, we will conduct and include an ablation study on the impact of shot numbers on model performance in our revised manuscript. For instance, in our experiments with LLaMA-3-8B, we observed accuracy rates of 0.3300, 0.3462, 0.3528, 0.3539, and 0.3536 for shot numbers 1, 2, 3, 4, and 5 respectively. These results suggest that increasing the number of shots generally improves performance, with diminishing returns after 4-5 shots.
>
> 5. Privacy issues and potential negative social impacts:
> We acknowledge that we should have addressed this aspect more explicitly. While our dataset does not involve personal or private information, we recognize the importance of discussing potential ethical implications. We will add a section in the revised manuscript to clarify the absence of privacy concerns and to address any potential ethical considerations or social impacts of our benchmark.

---

> > ### Comment · Reviewer_WZvx · 2024-08-22
> > **Response to authors - rebuttal**
> >
> > Thank your for providing some insights to the raised issues.
> > I'm happy with the promised modifications upon the acceptance and will keep my score.

---

### Official Review · Reviewer_ZrEH · 2024-08-03

**Rating:** 6
**Confidence:** 4
**Correctness:** Yes.
**Clarity:** Yes.

**Review:**

The pros of this paper:
1. Valuable alternative and enhanced benchmark for LLM.
2. Super comprehensive evaluation of LLM on the introduced dataset.
3. Well presented and easy to follow.
4. Inclusion of expert review.

I have the following concerns and questions:
1. Could the author provide more details about the expert review? Such as how many experts were involved and their qualifications. More transparency regarding the selection of experts, the criteria they used, and how disagreements were resolved would enhance the credibility of this process.
2. I have concerns about the fallback mechanism in “a fallback mechanism is implemented where a random option from the answer choices is selected” (line 195). Please provide evidence to show this operation is rational and their influence on the result. As the answer extraction is also very important for LLM evaluation.
3. Figure 4 and its analysis are misleading. 1) The ranking of this figure is based on the value of MMLU-pro, making MMLU an unfair position. As this experiment claims the accuracy span of MMLU-pro is larger than that of MMLU, thus the difference in accuracy is enough here.  2) The following claim is a cherry-pick actually. “For instance, models like Gemini-1.5-Flash, Llama-3-70B-Instruct, Phi-3-medium-4k-instruct, and Qwen1.5-110B all score between 78% and 82%, a narrow 4% range that encompasses four models, challenging the differentiation of their performance.” (line 253)

**Strengths:**

see Review

**Additional Feedback:**

None

**Documentation:**

Yes.

**Limitations:**

Yes.

**Opportunities For Improvement:**

see Review

**Relation To Prior Work:**

Yes.

**Summary And Contributions:**

This paper introduces MMLU-Pro, builds upon the widely-used MMLU benchmark but addresses its the following limitations by:

- Increasing the number of choices from four to ten: This reduces the chance of correct guessing and enhances the benchmark's difficulty.
- Incorporating more complex, reasoning-focused questions: This addresses the performance saturation observed with existing benchmarks and promotes CoT reasoning.
- Integrating expert reviews: This minimizes dataset noise and ensures higher quality questions.

Along with the dataset, this paper evaluates 50 LLMs, including both open-source and closed-source models, on MMLU-Pro. The resulsts demonstrate that MMLU-Pro successfully distinguishes model capabilities better than MMLU as claimed.

---

> ### Author Rebuttal · Authors · 2024-08-15
>
> Thank you for your thorough review and constructive feedback on our MMLU-Pro paper. We appreciate your recognition of the paper's strengths and would like to address your concerns:
>
> 1. Expert review details:
> We appreciate your request for more transparency. Our expert reviewers all have at least a bachelor's degree in STEM fields, with coursework covering the topics addressed in the questions. When uncertain about a question, reviewers consulted relevant materials and engaged in group discussions to reach a consensus. We will include more details about this process in the revised manuscript.
>
> 2. Fallback mechanism:
> Thank you for raising this important point. We'd like to clarify two aspects:
> 1）Firstly, the ability to follow instructions is a crucial evaluation metric for LLMs. We explicitly specified the response format in the prompt and few-shot demonstrations. When answers don't adhere to this format, we consider it reasonable and fair to use a random selection as a fallback. It is worth noting that most models do not trigger the "random selection" much. Its impact is rather small in most cases.
> 2）Secondly, for high-performing models like GPT-4, Claude-3.5, Gemini, etc, answer extraction failures are extremely rare. We discuss this in Section 5.3, where we provide an error analysis detailing the success rate of answer extraction under various conditions and the specific impact of the fallback strategy when failures occur.
>
> 3. Figure 4 and analysis:
> We appreciate your insight regarding the potential bias in our analysis. We agree that our current presentation could be improved. In the revised manuscript, we will remove the selective quoting and instead use the Coefficient of Variation to quantify the discriminative power of MMLU-Pro and MMLU. Among the models presented in Figure 4, the Coefficient of Variation for MMLU is 0.0878, whereas for MMLU-Pro, it is 0.2278. This more objectively demonstrates MMLU-Pro's enhanced ability to differentiate between model performances.
>
> Thank you again for your valuable feedback that helps us further improve paper quality.

---

### Official Review · Reviewer_AG1t · 2024-08-07
**The paper talks about creation of a dataset that is challenging and raises the bar for LLM benchmarks and it has removed noisy data.**

**Rating:** 7
**Confidence:** 3
**Correctness:** The claims made in the submission are…
**Clarity:** The paper is well written

**Review:**

The paper talks about limitations in existing MMLU dataset and how MMLU pro is better compared to the earlier.

**Strengths:**

There are several strengths in this paper, they are :-

1. Reduction in trivial questions and noise : MMLU-Pro introduces more challenging questions and increases the number of options, making it harder for models to guess answers.

2. Focus on reasoning : The benchmark emphasizes reasoning over mere knowledge recall, which is crucial for advancing AI capabilities.

4. Extensive Evaluation : The authors conducted thorough experiments with over 50 LLMs, providing a clear comparison between MMLU and MMLU-Pro.

5. Chain of Thought Reasoning: Highlighting the importance of CoT reasoning aligns well with current trends in AI research.

**Additional Feedback:**

No additional feedback

**Documentation:**

There is sufficient information related to the data

**Ethics:**

No ethical concerns

**Limitations:**

1. The dataset is extensively huge and very resource intensive.

**Opportunities For Improvement:**

1. The dataset could be extended to handle multiple languages to increase the scope

2. Some details about practical applications of MMLU pro would help.

**Relation To Prior Work:**

Yes it is clearly mentioned

**Summary And Contributions:**

The paper talks about creation of Multi Modal dataset which is different from general dataset. The main contribution is the creation of difficult questions and answers and removal of trivial questions which helps in evaluation and betterment of LLMs. The authors used chain of thought promoting and found that useful

---

> ### Author Rebuttal · Authors · 2024-08-15
>
> Thank you for your thoughtful review and positive feedback on MMLU-Pro. We appreciate your recognition of our work's strengths and the overall positive evaluation.
>
> Regarding your suggestions for improvement:
>
> 1. Multiple languages: We agree that extending the dataset to handle multiple languages would increase its scope and utility. We will consider this as an important direction for future work.
> 2. Practical applications: We appreciate this suggestion and will briefly discuss the potential practical applications of MMLU-Pro in the revised version of our paper.
>
> Concerning the limitation you mentioned about the dataset being extensively huge and resource-intensive:
>
> We understand this concern. However, we'd like to clarify that the computational cost for evaluating models on MMLU-Pro is not as high as it might seem. In our implementation, we utilized the vllm acceleration framework, significantly reducing resource requirements. For example, using a single A100 GPU, the inference time for completing the MMLU-Pro evaluation with an LLaMA-3-8B model is only about 50 minutes. We believe this demonstrates that MMLU-Pro balances being comprehensive and computationally feasible for most research teams.
>
> Thank you again for your valuable feedback. We look forward to incorporating your suggestions to further improve our work.

---

### Decision · Program_Chairs · 2024-09-26

**Decision:**

Accept (Spotlight)

**Comment:**

This paper introduces MMLU-Pro, an extension of MMLU with (1) more number of choices, from four to ten (2) incorporating more reasoning focused questions. (3) incorporating expert reviews (with at least BSc degree as the author clarified in the rebuttal). Experimental results shows that MMLU-Pro is more difficult than MMLU reducing the overall accuracy score by 16% to 33%.

Overall, all reviewers agree that this is a great contribution, and they suggested multiple areas of extension like extending it to other languages. I agree with reviewer WZvx to have a comparison table showing the differences between MMLU and MMLU-Pro.